# Offline replay supports planning in human reinforcement learning

Ida Momennejad[1]*, A Ross Otto[2], Nathaniel D Daw[1], Kenneth A Norman[1]

[1]Princeton Neuroscience Institute, Princeton University, New Jersey, United States; [2]Department of Psychology, McGill University, Montreal, Canada

**Abstract** Making decisions in sequentially structured tasks requires integrating distally acquired information. The extensive computational cost of such integration challenges planning methods that integrate online, at decision time. Furthermore, it remains unclear whether 'offline' integration during replay supports planning, and if so which memories should be replayed. Inspired by machine learning, we propose that (a) offline replay of trajectories facilitates integrating representations that guide decisions, and (b) unsigned prediction errors (uncertainty) trigger such integrative replay. We designed a 2-step revaluation task for fMRI, whereby participants needed to integrate changes in rewards with past knowledge to optimally replan decisions. As predicted, we found that (a) multi-voxel pattern evidence for off-task replay predicts subsequent replanning; (b) neural sensitivity to uncertainty predicts subsequent replay and replanning; (c) off-task hippocampus and anterior cingulate activity increase when revaluation is required. These findings elucidate how the brain leverages offline mechanisms in planning and goal-directed behavior under uncertainty.
DOI: https://doi.org/10.7554/eLife.32548.001

## Introduction

To make choices in structured environments, human and non-human animals must forecast the long-term consequences of candidate actions, which often are delayed and play out serially over multiple steps. Predicting these outcomes often requires integrating information from disparate learning episodes, which occurred at different times in the past. In other words, successful planning requires both (a) accessing memories of multiple previous events and (b) integrating these memories to make useful predictions about an action's consequences: a type of integrative memory replay (*Lengyel and Dayan, 2008*). Indeed, both human and non-human animals can solve reward learning tasks designed to require this sort of integration, such as latent learning tasks (*Tolman, 1948*) in which information about the outcome contingencies of actions (e.g., the spatial layout of a maze) is presented in a separate stage of training from the associated rewards (e.g. receipt of food).

Little is known, however, about how and when such integrative planning occurs. A longstanding assumption, dating back to Tolman's proposal in 1948 (*Tolman, 1948*), is that planning unfolds at decision time, for example during mental simulation of the candidate future action trajectories. More recent proposals suggest that such planning need not occur only when the animal is presented with a choice, but can also occur earlier, during *encoding* of choice-relevant information (*Wimmer and Shohamy, 2012*; *Shohamy and Daw, 2015*). In particular, when new information is first learned (e.g. food discovered in some location), its consequences for distal actions (e.g. a high value for routes leading there) can be inferred and immediately encoded. Indeed, human fMRI work provides support for both mechanisms: Studies have found that people's ability to solve integrative evaluation problems depends on memories of distal task states accessed either at the time of choice (*Doll et al., 2015*) or encoding (*Wimmer and Shohamy, 2012*).

What the encoding- and choice-time mechanisms share is that both occur during active task performance, which artificial intelligence refers to as learning *online* (*Sutton, 1991*). However, during

*For correspondence:
idam@princeton.edu

Competing interests: The authors declare that no competing interests exist.

both encoding/learning and decision-making periods, people often have limited time for exploring the set of interrelationships among decisions and outcomes, which may be prohibitive in complex environments. A third, more general possibility, which we examine here, is that integration of memories to support action evaluation can also occur *offline*, such as during off-task rest periods or even sleep. An advantage of offline planning is that it offloads computational costs to periods that are not already occupied by task behavior. This way, before the organism faces a problem that requires a decision (but after it has initially encoded the constituent events), it can update action valuations or choice policies offline, thereby accomplishing at least part of the evaluation required for later choice (*Gershman et al., 2014*).

Here we investigate integrative memory processes in humans that help piece together information offline, thus supporting planning and improving decision-making behavior. In the formalism of reinforcement learning in artificial intelligence, the *Dyna* family of algorithms implements offline planning, which is accomplished by learning from *replay* of experiences that were initially encoded during online behavior (*Van Seijen and Sutton, 2015*; *Sutton, 1991*). Inspired by this work, and together with behavioral evidence of offline revaluation (*Gershman et al., 2014*; *Momennejad et al., 2017*), computational models suggest that *Dyna*-like offline replay mechanisms might be implemented in the brain (*Momennejad et al., 2017*; *Johnson and Redish, 2005*; *Ludvig et al., 2017*; *Russek et al., 2017*). Indeed, this hypothesis fits closely with a substantial rodent literature revealing that hippocampal replay of previously experienced spatial trajectories is observed during both awake behavior and sleep (*Takahashi, 2015*; *Ólafsdóttir et al., 2015*).

It has also been shown that hippocampal replay (*Pfeiffer and Foster, 2013*) captures the topological structure of the task environment (*Wu and Foster, 2014*), and hippocampal reverse replay of trajectories occurs offline when the animal experiences changes in reward (prediction errors) at the goal state (*Ambrose et al., 2016a*). In parallel, human fMRI research work reveals how memories are reactivated offline, such as during periods of rest that follow rewarding episodes (*Gruber et al., 2016*; *Murty et al., 2016*). This offline reactivation could potentially aid in memory consolidation and integration, evidenced by enhanced subsequent memory performance (*Takashima et al., 2009*; *Staresina et al., 2013*; *Tambini and Davachi, 2013*). Consistent with the hypothesis that memory integration can be accomplished via offline replay, there is behavioral evidence that manipulations affecting rest periods influence humans' ability to solve integrative decision tasks (*Gershman et al., 2014*). However, so far, no animal or human study has provided neural evidence that integration during offline replay supports planning.

Our main question in this study is: does replay predict a change in choice behavior? In this study, we use fMRI and multivariate pattern analysis (MVPA) to test the hypothesis, outlined above, that offline replay drives integrative re-evaluation and update of past policies. We designed a paradigm where such integration would be reflected in the behavioral choices participants make. It has been shown that replay of a particular episode can lead to the reinstatement of neural activity in sensory cortical regions (*Ji and Wilson, 2007*; *Polyn et al., 2005*). Accordingly, we use MVPA here to track cortical reinstatement of a particular state during off-task rest periods, as a measure of offline replay of that state. We then examine the relationship between neural replay evidence (from MVPA) and the change in subsequent choice behavior, to test whether replay facilitates the brain's ability to integrate information and update planning policies accordingly.

Having found evidence for this relationship between replay and replanning, our secondary question in this study was to explore the antecedents of this process: What determines which memories are replayed? The idea that integrative replay occurs offline (rather than being triggered by particular task events) raises the question of *prioritization*: Out of all the episodes stored, which memories should the brain replay to ensure integrative and goal-directed learning? What is it about an experience that privileges it for later replay? Here we draw insight again from artificial intelligence, where the standard approach for this problem is an algorithm known as prioritized sweeping (*Moore and Atkeson, 1993*; *Peng and Williams, 1993*). The key insight behind this algorithm is that when surprising information is encountered at some state, it may have consequences for the value of states or actions that could have led there, and therefore those predecessors should be prioritized for replay. One representative and standard signal of such surprise or uncertainty, which is leveraged as a heuristic to trigger replay in prioritized sweeping, is the experience of prediction errors—that is the deviation between the agent's predictions and experiences of the world. To this end, both negative and positive prediction errors (i.e. the unsigned error) are meaningful as they signify the need

for improved value estimates. To test this hypothesis, we use model-based fMRI analysis to evaluate whether, across blocks and subjects, larger neural correlates of prediction errors are associated with greater replay during subsequent rest periods, and ultimately greater change in behavior.

In short, our major proposal is that offline neural replay drives the integration of distal memories to update previously learned representations and support future planning. Our secondary question is which earlier events predict replay and ultimately replanning. We hypothesize that episodes marked with large prediction errors (PEs), whether positive or negative, signal uncertainty about the optimal reward trajectory and tag related memories for future replay. This replay, in turn, enables the integration of separate memories in the service of updating optimal reward policy (action policies that maximize reward). We combine model-based fMRI, MVPA, and a sequential decision-making task to test our hypotheses.

## Results

To test the hypothesis that offline replay supports integration of memories to serve future decisions, we operationalized planning using a reward revaluation paradigm (*Gershman et al., 2014*; *Momennejad et al., 2017*). In reward revaluation, participants first learn the multi-step sequential decisions that lead them to reward from a starting point; later, they experience a local change to later stages of the decision trajectory. Upon return to the beginning of the trajectory, we can measure the extent to which a participant's choices adapt to the new local changes they experienced.

Here we used a 2-stage sequential decision task with a within-subject design that allowed us to manipulate whether or not revaluation was required for reward-maximizing choice behavior (*Gershman et al., 2014*; *Momennejad et al., 2017*). In both the revaluation and control conditions, participants (*n* = 24) repeatedly visited a starting stimulus (Stage I) and chose between two actions that would lead them to one of two subsequent stimuli depending on their choice (Stage II), as displayed in *Figure 1*. Once a participant reached either of those Stage II stimuli (or states), they selected one of two actions to obtain reward. We will refer to the Stage I stimulus as *state one* and the two Stage II stimuli as *state 2* and *state 3*. As a cover story, participants were told they were playing a game as a thief, exploring different buildings to steal the most money. Participants were given four separate blocks of this task, each with a different reward structure; half these blocks were randomly assigned to the *revaluation* condition, and half the blocks were assigned to the *control* condition, as described below. To ensure that participants did not transfer what they learned from one block to another, each block had a different colored background that indicated they were in a different city, and different stimuli for the Stage I and Stage II actions (*Figure 1—figure supplement 1*).

The task in both revaluation and control blocks consisted of three phases: a Learning phase (where the participant explored the entire decision tree), a Relearning phase (in which the participant visited Stage II but never Stage I), and a Test phase (where participants revisited Stage I). During the Learning phase, participants learned the structure of states and rewards.

During the Relearning phase, the rewards associated with Stage II stimuli changed for the revaluation blocks (but not for the control blocks). The new reward amounts in revaluation blocks were chosen so as to require a change in the decision policy leading to optimal reward in Stage I (e.g., during the Learning phase, the optimal decision policy might be to go left at Stage I, but after taking into account the Stage II reward changes experienced during Relearning, the new optimal decision policy might be to go right at Stage I).

The Test phase assayed whether participants reversed their initial preference according to the Stage II changes they experienced in the Relearning phase (in control blocks we did not expect any change in preference to occur). Note that a change in Stage I policy preference would require integrating information about the action-stimulus contingencies from the Learning phase with the relearned reward amounts associated with Stage II states from the Relearning phase. The Test phase choice was repeated four times; we computed the average proportion correct across these four choices. Importantly (to prevent any influence of new learning during the test phase), participants were not given any feedback about the consequent Stage II stimulus and reward after these choices.

Participants were incentivized to search for and choose higher-value states. They were explicitly instructed prior to the learning phase to find the highest monetary reward and navigate there in order to increase their bonus money. The instructions below appeared prior to the learning phase of

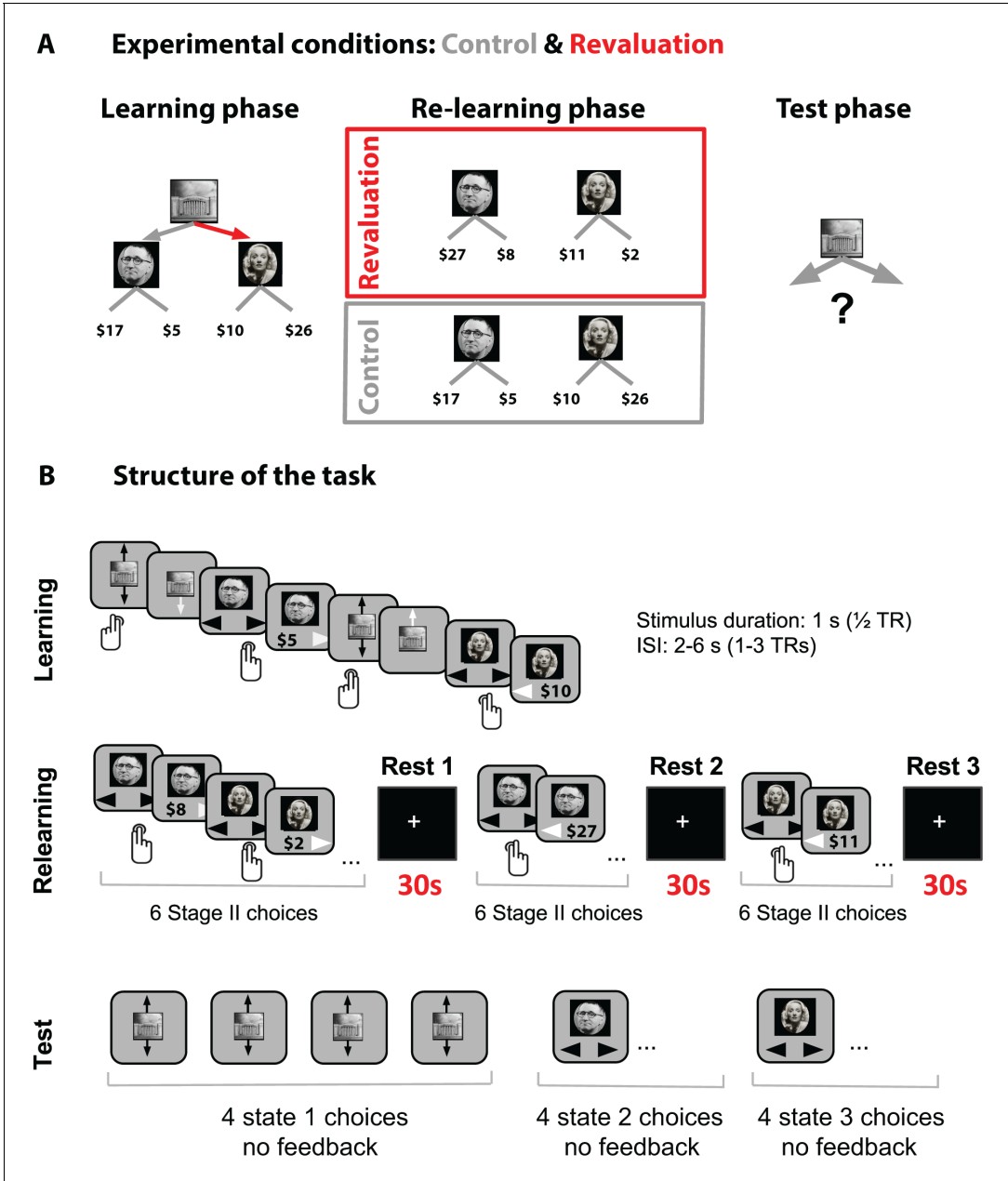

**Figure 1.** (A) Task design Each block had three phases. During the Learning phase, participants explored a 2-stage environment via button presses to learn optimal reward policy. During the Relearning phase, they only explored the Stage II states: on revaluation trials the optimally rewarding Stage II state changed during the Relearning phase; on control blocks the mean rewards did not change. The final phase was the Test phase: participants were asked to choose an action from the starting (Stage I) state that would lead to maximal reward. The red arrow denotes the action that provides access to the highest reward in the learning phase. The optimal policy during the Learning phase remains optimal in the control condition but is the suboptimal choice in the revaluation condition. (B) The time course of an example block. Stimuli images were downloaded from the public domain via Google images.

DOI: https://doi.org/10.7554/eLife.32548.002

The following figure supplement is available for figure 1:

**Figure supplement 1.** The four different blocks.

DOI: https://doi.org/10.7554/eLife.32548.003

every city/MDP during the experiment in order to reiterate this incentive (the names of the city and location were adjusted for each session):

"In this game you will be robbing a theater in Berlin.

By learning how to get around the theater, and learning where the most money is, you can steal the most money possible.

And the more money you steal in the game, the more REAL BONUS MONEY you will be paid on top of the normal experiment payment. You could earn an extra $2–3!'

Note that the critical difference between the control and revaluation conditions was in the Relearning phase: in the control condition there was no change in the rewards associated with Stage II choices (such that integrative consideration of this new information should not result in systematic change in Stage I preferences), whereas in the revaluation condition Stage II rewards changed. Hence, during the Test phase we could test whether decisions made in Stage I were affected by information learned about Stage II in the Relearning phase (see *Figure 1* and Materials and methods). The inclusion of the control condition helps to rule out effects related to any nonspecific change in preferences (as opposed to integrative replanning), such as forgetting or inattention. As noted above, participants completed four separate blocks with this structure but different stimuli and rewards (two revaluation and two control). They did not know to which condition each block belonged (or about the logic of the manipulation), and the order of revaluation vs. control blocks was randomized.

In addition to the revaluation vs. control manipulation, we also manipulated the level of noise (variance) associated with rewards experienced following an action in a given state. This led to a 2 × 2 factorial design with revaluation and noise manipulations. For both revaluation and control blocks, half the blocks had variable (noisy) rewards selected from a distribution with a fixed mean (*noisy* rewards condition) leading to a persistent level of prediction errors even in the control condition (e.g. choosing left at state two would lead to different values that average to $5), and half of the blocks had no noise in rewards (*noiseless* rewards condition) leading to no prediction errors after learning unless there was a revaluation (e.g. choosing left at state two would always lead to $5).

This noise manipulation—that is the noiseless-rewards versus noisy-rewards conditions—ensured that the difference between revaluation and control condition blocks was not limited to the absence or presence of PEs or surprise in general. In particular, the inclusion of a noisy control block (in which there is elevated surprise but no actionable new information during the Relearning phase) further helps to control for nonspecific changes in preference (e.g., mediated by attention or motivation) that might be associated with surprise per se, rather than surprise enabling integrative planning. Note that under the Dyna and prioritized sweeping algorithms that underlie our hypotheses, replay by itself is not sufficient to produce change in preference. Our hypothesis, instead, is that surprise leads to replay later on (to investigate whether the surprising information has consequences for choice), but whether this replay leads to replanning/revaluation depends on whether the replayed experiences in fact support a change in preference. Accordingly, the inclusion of reward noise in the noisy-rewards condition (even in control blocks) allowed us to rule out the possibility that change in choice preferences is brought about by 'any' experience of uncertainty, whether or not experience actually supported replanning. However, given that our primary interest is in replanning per se, in what follows we mainly report revaluation vs. control effects, collapsing noiseless and noisy reward conditions, while reporting statistical differences and further fMRI analyses of the noise manipulation in the Supplementary material.

## Instructions

Below is an example of instructions for one trial. Each trial had a separate city context (Berlin, Paris, London, Kyoto) and place (Theater, Museum, Subway, Garden).

"In this game you will be robbing a GARDEN in KYOTO.

By learning how to get around the GARDEN, and learning where the most money is, you can steal the most money possible.

And the more money you steal in the game, the more REAL BONUS MONEY you will be paid on top of the normal experiment payment. You could earn an extra $2–3!'

When you are in the Main KYOTO GARDEN, you either go to the LEFT or the RIGHT garden to find money. The MAIN GARDEN and the LEFT and RIGHT gardens all have different PHOTOS, so you will always know where you are.

Pay attention! Sometimes you will directly begin in the LEFT or in the RIGHT garden!

Once you enter the LEFT or RIGHT garden, you will have the choice of 2 cash registers to look for money in: TOP or BOTTOM machines.

Each machine contains between 1$ and 50$.

You need to figure out whether the TOP or the BOTTOM machine pays better in each garden.

Stealing as much money as possible requires that you successfully navigate from the MAIN garden LEFT or RIGHT to the GARDEN with the best cash register.'...

Beware!! Press THUMB if you find yourself in a suspicious corridor where you may get caught!'

## Forced choice catch stimuli

It is worth noting that every trial consisted of a number of catch stimuli. These 'catch states' were included to control for attention and learning. There were two kinds of catch states: forced choice and police states. In forced choice catch states, participants faced the same choice but one of the actions was highlighted, thus they were forced to select the indicated response. Forced choices were included in the design in order to ensure (a) that they sample all parts of the state space and (b) that they are paying attention. On 'police' states, upon choosing an action for State 1, participants saw a 'police face' stimulus (if the second state stimuli were faces) or a 'suspicious corridor' stimulus (if the second state stimuli were scenes), which implied that they should not press the usual response and they need to press an extra key (to avoid being 'caught'). Participants were trained and could distinguish police faces and suspicious corridors beforehand.

The number of forced choice changed depending on the performance of the participant. Including forced-choice trials ensured that the distributions were adequately sampled during the experiment. Each participant's choices were tracked, and actions that were not adequately sampled were presented in forced choice trials in which the participant had to take those actions. We included 4 'police' catch trials and, on average, 3 'forced choice' catch trials in phase 1 of each run. We included at least 1 of each less explored option. If the participant did not adequately explore options within a set period of time, more catch stimuli were inserted just to ensure participants encountered the value of those states. As a result of forced choice stimuli, participants sampled all options. As a result of police trials, we ensured that they did not merely rely on stimulus-independent motor sequence memory when making decisions (e.g. memorizing that pressing L-index then R-middle is rewarding). Instead, participants were directed to pay attention to the stimuli/states they observed in order to select the optimal action. They were told that these catch choices are important for their bonus. Note that we ensured all varieties of catch trials were category-congruent with the second stage stimuli.

## Attention engagement during the relearning (control trials)

Participants were instructed at the beginning of each run that the rewarding choices may change and they needed to pay attention to changes. Furthermore, both the instructions as well as forced choice stimuli incentivized them to pay attention during the relearning phase: (a) The relearning phase had 'forced choice' trials where participants were instructed to select the option already highlighted by the experimenter. We used these trials to ensure they were paying attention and provided verbal feedback between the runs to pay attention to these forced choices. (b) Participants were told their choices during the relearning phase were also related to reward at the end of the experiment (maximizing rewards required maximizing performance as a 'thief' in the cover story). (c) Participants were sometimes given catch trials with 'police' stimuli that required a different response, as noted above. Taken together, all of these factors helped to ensure that participants were paying attention in both the revaluation and the control condition. Note also that participants were not directly told whether a particular block was a revaluation block or control block – they had to pay attention to experienced rewards in all blocks during relearning to assess whether they were changing.

Moreover, we designed the noisy control condition to control for the possibility that participants were not paying enough attention to details and simply followed a rule of thumb: 'any change means revaluation'. Our observations ruled out this possibility: the mere experience of PEs did not lead participants to change their choice at the end of noisy control trials. Note that even in the relearning phase of the noisy control condition there were still non zero PEs, indicating that

participants were still learning and improving their predictions. They did not know they were in the control condition and were incentivized to pay attention.

Our main neural hypothesis was that offline replay would facilitate the integration of Stage I and Stage II information, which will in turn support subsequent planning behavior, that is, change in choice preferences. In the control condition, participants should not change their choices from learning to test regardless of experienced PEs, while in the revaluation condition, replay should help them update previous optimal policy and replan their choice. Note that updating or replanning the optimal Stage I choice required integration of past information, because the transition from Stage I to Stage II, on the one hand, and the new Stage II rewards, on the other, were originally experienced at separate distal points in time during online task performance.

In our study, the Relearning phase of each block was interleaved with three separate rest periods of 30 s each. Thus, offline replay could be assessed using brain measurements during these off-task rest periods. To allow us to measure replay, we used image categories for Stage I and Stage II stimuli that were easy to decode with fMRI: All stimuli were either faces or scenes (*Polyn et al., 2005*). In particular, the Stage I (root) stimulus belonged to a different category than the two Stage II stimuli. This manipulation allowed us to operationalize 'offline replay' using classifier evidence of Stage I in separate category-selective regions of interest (ROI) during rest periods, as described below.

## Replanning (revaluation) behavior

To measure revaluation behavior, we computed a *replanning magnitude* score for each subject and block (*Gershman et al., 2014*; *Momennejad et al., 2017*; *Russek et al., 2017*), defined as the proportion of optimal Stage I choices during Test minus the proportion of choices for that same option during the Learning phase (see *Equation 1* in Materials and methods). We use a score that indexes *change* in preferences here (*Gershman et al., 2014*), rather than raw Test phase choice behavior for two reasons. First, this is what our hypotheses concern: that replay updates pre-existing choice policies, driving a change. Second, this score controls for any confounding effects of initial learning performance on Test phase behavior. (For instance, subjects paying different amounts of attention might learn to varying degrees in the initial phases, leading to differences in both replay and Test phase behavior, even if replay itself doesn't affect later preferences.) In revaluation blocks, this replanning magnitude quantifies successful reversal of preference from Learning to Test, because the initially suboptimal option becomes optimal during the Test phase. The difference in preference captured by the replanning magnitude compares the frequency of taking the newly optimal action during the Test phase to the frequency of taking this action at the end of the Learning phase (when it was suboptimal), which adjusts for choice noisiness or failure to acquire the initial associations during the Learning phase. Thus, for revaluation blocks, a large replanning magnitude indicates consistent, successful reversal of an initially consistent, correct preference. Conversely, in control blocks (where the initially suboptimal option remains suboptimal at test) a large replanning magnitude indicates an unwarranted and suboptimal reversal of the initially trained preference, allowing us to control for preference change due to some other, nonspecific source of error such as decision noise or forgetting.

## Behavioral results

Comparison of replanning magnitudes revealed that participants showed significant revaluation behavior (i.e., reversed their Stage I choice from Learning to Test, as indicated by positive replanning magnitudes) in revaluation blocks ($t(23) = 4.89$, $p < 0.0001$), but not during control blocks ($t(23) = -0.10$, $p = 0.92$), and the difference in magnitudes between the two conditions was significant ($t(23) = 4.333$, p = 0.0002) (*Figure 2B*). The comparison with the control condition verifies that this apparent revaluation behavior is not simply attributable to general forgetting or increased memory noise from Learning to the Test phase; rather, it is due to revaluation of past policies in the face of novel information in the revaluation condition—but not the control condition. A 2-way analysis of variance confirmed the main effect of the reward vs. control factor but revealed no significant main effect of the reward stability vs. noise manipulation (see *Figure 2—figure supplement 1*).

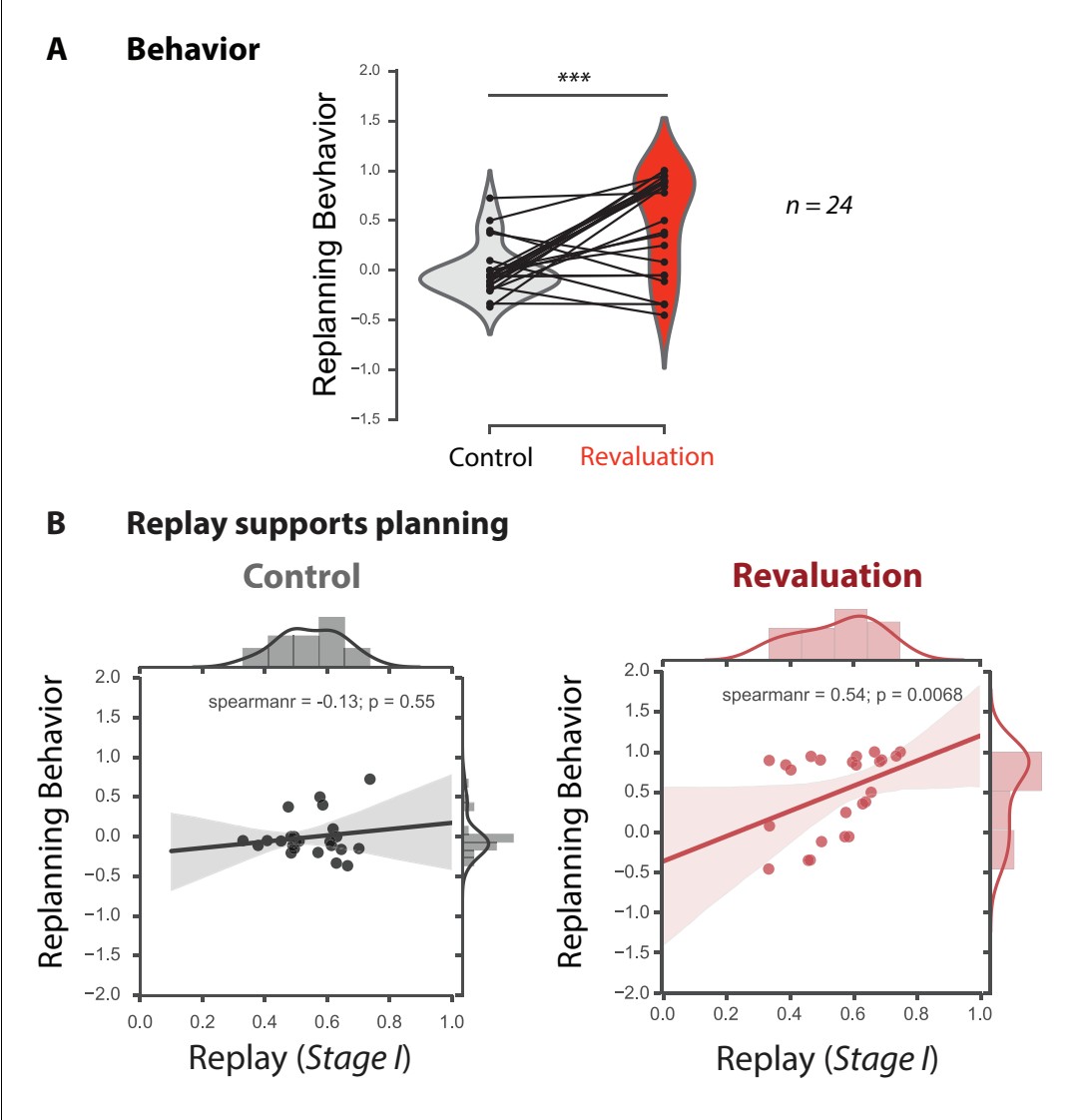

**Figure 2.** Offline replay of distal past states supports planning. (**A**) Behavioral results (n = 24). Participants significantly changed their choice from the Learning phase to the Test phase in the revaluation condition, but not the control condition. (**B**) Replay supports planning. Correlation between Stage I replay during off-task rest periods and replanning magnitude during the subsequent Test phase for Control (left) and Revaluation (right) blocks. Stage I replay was operationalized as MVPA evidence for the Stage I stimulus category, in category-selective regions of interest, during all rest periods of control and revaluation conditions (n = 24). The correlation was significant in the Revaluation blocks (Spearman rho = 0.54, p = 0.0068) but not Control blocks (Spearman rho = −0.13, p = 0.55), and it was significantly larger in Revaluation than Control blocks (p = 0.0230, computed using a bootstrap, computing Spearman rho 1000 times with replacement). Regression lines are provided for visualization purposes, but statistics were done on Spearman rho values. The bottom figures display both fitted lines as well as Spearman's rho values for clarity.
DOI: https://doi.org/10.7554/eLife.32548.004

The following figure supplements are available for figure 2:

**Figure supplement 1.** (**A**) Illustration of experimental conditions obtained by crossing revaluation and noise.
DOI: https://doi.org/10.7554/eLife.32548.005

**Figure supplement 2.** We tested whether Stage two replay was also correlated with subsequent planning (revaluation) behavior.
DOI: https://doi.org/10.7554/eLife.32548.006

## Replay-behavior relationships

Next, we turned to the neuroimaging data to test our main hypothesis: whether neural evidence for offline replay during rest periods predicts subsequent replanning behavior. As noted above, we used MVPA to track replay of Stage I, leveraging the fact that Stage I and Stage II stimuli were

always taken from different categories (faces and scenes). For example, in a trial where the Stage I stimulus was a face and Stage II stimuli were scenes, we used MVPA measures of face activity during rest periods to measure spontaneous replay of Stage I. Previous evidence suggests that multivariate pattern analysis (MVPA) can be used to measure offline replay and memory reactivation during post-task rest periods (*Staresina et al., 2013*; *Cichy et al., 2012*; *Gruber et al., 2016*). Specifically, it has been shown that imagery, reinstatement, and perception of faces and scenes elicit patterns in the temporal cortical regions associated with visual categories, roughly corresponding to the fusiform gyrus for the face category and the parahippocampal gyrus for the scene category (*Polyn et al., 2005*; *Eichenbaum, 2004*). Therefore, we first identified these category-selective regions using general linear models (GLMs) on a separate localizer run before the main task. The localizer run consisted of blocks of faces, scenes, objects, and rest. The faces and scenes, but not objects, were used later during the experiment. We then trained a classifier (using logistic regression) on face, scene, and object stimuli appearing during the separate localizer run. The main goal of the classifier was to classify the reinstatement of cortical patterns associated with the Stage I stimulus category during the rest periods. As a cortical measure of replay, for each participant, each block, and each of the 30 s rest periods, we used this logistic regression classification method to identify TR-by-TR classifier evidence for the cortical reinstatement of faces and scenes in category ROIs (regions of interest). To ensure that we could independently assess positive evidence for both face and scene categories (rather than competitively detect one at the expense of the other), we assessed classifier evidence for each category from an ROI that was selective for that category (see Methods for further discussion of this point).

To test whether offline replay during intermittent rest periods in the Relearning phase predicted successful revaluation in the later Test phase, we averaged MVPA evidence for reinstatement of the Stage I category during all rest periods (*Figure 2*, x-axis), and computed the correlation with mean revaluation behavior in each condition (*Figure 2*, y-axis), across subjects. Even though the Stage I stimulus category never appeared during the Relearning phase, it was important to ensure that we were not simply observing residual signals evoked by the Stage II stimuli observed prior to each rest phase. Therefore, we excluded the first 5 TRs (corresponding to 10 s) of the beginning of each rest period from analysis.

In revaluation trials, there was a significant correlation between MVPA evidence for Stage I replay during the three off-task rest periods and revaluation behavior during the subsequent Test phase (*rho = 0.54, p = 0.0068*) but not for control trials (*rho = −0.13, p = 0.55, Figure 2*). The correlation coefficients for the revaluation vs. the control trials were significantly different from one another (p = 0.0230; bootstrap, see Methods). We also conducted the same analyses, separately, for the noisy rewards versus the noiseless rewards conditions. The revaluation vs. control difference was trending in the noisy condition (p = 0.066) but not the noiseless condition (p = 0.16), and there was no significant interaction between revaluation/control and noisy/noiseless (p = 0.34) (*Figure 2—figure supplement 1*; see Discussion section for further treatment of these results).

Note that the crucial step for revaluation is retrieval of memory for the Stage-1 state, which was not presented in the Relearning phase. In contrast, there may be neural activity for the Stage-2 stimulus (which was presented during the Relearning phase) for many reasons unrelated to planning, such as simple working memory. Accordingly we expect the relationship between replay and revaluation to be most clearly expressed for the Stage one stimulus. We thus conducted a similar analysis for Stage two replay, looking at whether MVPA evidence for Stage two category replay during the rest period was correlated with subsequent replanning behavior. As expected, we observed no significant correlation between Stage two replay and revaluation behavior (*Figure 2—figure supplement 2*) in the revaluation condition (*rho = 0.2, p = 0.17*) or the control condition (*rho = -0.25, p = 0.22*).

In short, we found that MVPA evidence of Stage I replay during rest correlates with subsequent replanning behavior. This fits with our hypothesis that offline memory processes can support planning. Next, we test the hypothesis that this replay is driven by the experience of prediction error.

## The brain's response to unsigned prediction error tags memories for replay

A key question for any account of learning from replay is how the brain selects which memories to replay. We thus explored what events favor replay, and ultimately successful revaluation. A general,

appealing possibility is that events with some particular property when first experienced, like surprising ones, are privileged for later offline replay. (Mechanistically, this might occur either because they are 'tagged' for replay when first experienced, or because they are singled out later at retrieval.) We tested this idea in terms of one representative notion of surprise – neural sensitivity to unsigned PEs (i.e., the absolute value of PEs), as in the prioritized sweeping algorithm (*Moore and Atkeson, 1993*). Offline replay in turn would support updating past policies and hence predict future revaluation behavior (*Figure 3*, note the relationships indicated in the diagram, top left). We have shown in previous sections that offline replay during rest is correlated with revaluation during a later Test phase. To examine evidence for the hypothesized relationship between the brain's response to PE and offline replay, as well as subsequent replanning behavior (*Figure 3*, top left), we ran parametric modulation analyses as follows.

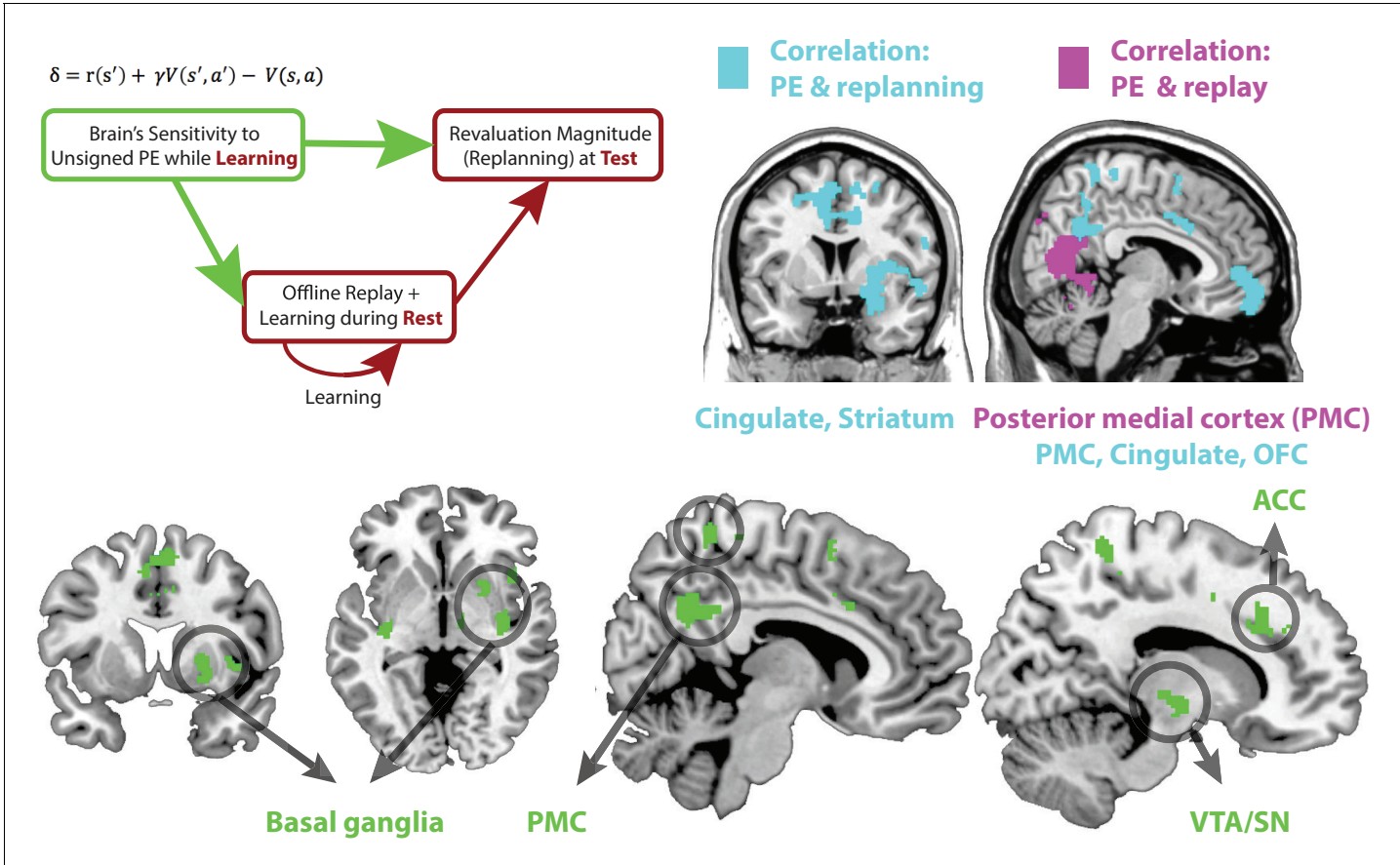

**Figure 3.** (Top left) Schematic of a theoretical account of revaluation. We propose that neural sensitivity to reward prediction errors during learning 'tags' or 'prioritizes' memories for replay during later rest periods. Replay during rest, in turn, allows the comparison of past policy with new simulated policy and updating of the past policy when needed. (Top right) Regions where sensitivity to unsigned PE in Revaluation blocks correlates with subsequent replay during rest (extent threshold p < 0.005, cluster family-wise error [FWE] corrected, p < 0.05, shown in purple) and revaluation behavior (extent threshold p < 0.005, cluster FWE corrected, p < 0.05, shown in blue). (Bottom) Green reveals the conjunction of regions where sensitivity to unsigned PE in Revaluation blocks correlates with subsequent replay during rest (extent threshold p < 0.005, cluster FWE corrected, p < 0.05) and revaluation behavior (threshold p < 0.005, cluster FWE corrected, p < 0.05) in those blocks; the conjunction is shown at a p < 0.05 threshold. We found that the sensitivity of broad regions in the basal ganglia, the cingulate cortex (including the ACC), and the posterior medial cortex (precuneus) to unsigned prediction errors (signaling increase in uncertainty) correlated with both future replay during rest as well as subsequent revaluation behavior. See *Tables 1* and *2* for coordinates.

DOI: https://doi.org/10.7554/eLife.32548.007

The following figure supplement is available for figure 3:

**Figure supplement 1.** Correlation between the brain's response to unsigned prediction errors and subsequent replay (purple), replanning behavior (blue), and their conjunction (green) in revaluation blocks, run separately for the noisy and noiseless conditions.

DOI: https://doi.org/10.7554/eLife.32548.008

For each participant, we conducted a parametric regression analysis within the Revaluation condition to identify regions with sensitivity to stimulus-by-stimulus unsigned PEs. Prediction errors were estimated from a Temporal Difference (TD) learning model (see Methods). We then tested, for each subject and each Revaluation block, the relationship between the magnitude of the brain's PE response (in that block) and two different covariates: subsequent replay and subsequent replanning score (*Figure 3*, top right, purple and blue maps respectively). This allowed us to determine whether (and in which brain areas) the magnitude of activity in response to unsigned PE was related to offline replay (*Figure 3*, regions indicated in purple, see *Table 1* for coordinates) and subsequent replanning (*Figure 3*, regions indicated in Blue, see *Table 2* for coordinates), as hypothesized (*Figure 3*, top left). In order to identify the combined effect and given the correlation between the two covariates, we used a conjunction analysis over the two resulting correlation brain maps (*Figure 3*, regions indicated in green). This conjunction analysis yielded regions where the brain's response to unsigned PE on each block correlated with both (a) subsequent mean replay evidence (for Stage I stimuli) during rest, and (b) subsequent replanning magnitude of the choice behavior during that block's Test phase (*Figure 3*).

This conjunction analysis yielded multiple regions where neural sensitivity to unsigned PEs significantly predicted both future offline replay and revaluation magnitudes: the anterior cingulate cortex (ACC), the mid cingulate, posterior medial cortex (PMC) including dorsal and ventral precuneus, and the basal ganglia, especially the putamen (*Figure 3*, bottom). These results are consistent with the hypothesized relationship between sensitivity to prediction errors during learning, offline replay during rest, and revaluation behavior at test, and suggest an involvement of ACC, PMC, and basal ganglia in signaling the relevant PEs (see *Figure 3—figure supplement 1* for breakdown of this analysis for the noisy-rewards and noiseless-rewards blocks).

## Brain activity during revaluation vs. control rest

Next, we report a number of observations further unpacking the key results discussed above. Our hypothesis is that learning and updating take place during offline replay periods—indeed, we revealed above that replay in category-sensitive brain regions during rest periods of revaluation blocks (but not control blocks) predicted subsequent planning behavior. This hypothesis also implies that brain regions related to learning and memory should show more activity during the rest periods of revaluation trials, where engaging in offline replay could result in better choices, compared to control trials in which the most learning has already occurred during the Learning phase. To test this, we used a GLM to compare overall differences in univariate activity during the rest periods of revaluation vs. control trials. Contrasting rest periods from revaluation trials versus control trials ($Rest_{revaluation} > Rest_{control}$) revealed higher activity in the hippocampus/medial temporal lobe and the anterior cingulate cortex ($p < 0.005$, cluster corrected at $p < 0.05$, *Figure 4*, see *Table 3* for coordinates, and *Figure 4—figure supplement 1* for breakdown of this analysis for the noisy-rewards and noiseless-rewards blocks).

## Learning during rest

So far, we have shown that evidence for offline replay, across subjects and blocks, correlated with subsequent replanning behavior, and that regions broadly involved in memory and evaluation were more active during the rest period of revaluation trials. Finally, we broke down these effects over the

**Table 1.** Coordinates of voxels where parametric modulation with unsigned PEs during learning predicted future replay of Stage I during rest periods, extent threshold $p < 0.005$, corrected at cluster level family-wise error $p < 0.05$ (these correspond to purple regions in *Figure 3*).

| Region | Z score | X | Y | Z | K voxels | P |
|---|---|---|---|---|---|---|
| Right Cuneus | 3.31 | -2 | −68 | 4 | 1898 | .000488 |
| Left Cuneus | 3.29 | 4 | −66 | 22 | | |
| Right Lingual | 3.21 | −14 | −68 | 22 | | |
| Left Lingual | 3.15 | 16 | −46 | -8 | | |
| Calcarine | 3.14 | 6 | −70 | 4 | | |

DOI: https://doi.org/10.7554/eLife.32548.009

**Table 2.** Coordinates of voxels where parametric modulation with unsigned PEs during learning predicted future replanning behavior (revaluation magnitude), extent threshold p < 0.005, corrected at cluster level family-wise error p < 0.05 (these correspond to blue regions in *Figure 3*).

| Region | Z score | X | Y | Z | K voxels | P |
|---|---|---|---|---|---|---|
| Right precentral | 3.67 | 26 | −20 | 68 | 5240 | 1.36e-08 |
| Orbitofrontal cortex (OFC) | 3.62 | 0 | 58 | −12 | | |
| Supplementary motor area | 3.59 | -8 | 4 | 48 | | |
| Right superior frontal cortex | 3.49 | 18 | 32 | 38 | | |
| ACC | 3.49 | −14 | 50 | 0 | | |
| Right OFC | 3.42 | 6 | 58 | −18 | | |
| Right superior temporal | 3.67 | 56 | −30 | 14 | 5165 | 1.67e-08 |
| Right supra-marginal | 3.6 | 34 | −36 | 44 | | |
| Inferior parietal | 3.58 | 36 | −42 | 44 | | |
| Right putamen | 3.39 | 26 | 8 | 2 | | |
| Right superior temporal pole | 3.35 | 46 | 18 | −16 | | |
| Left superior temporal pole | 3.4 | −46 | −10 | 0 | 2161 | .00021 |
| Left putamen | 3.35 | −20 | 18 | 8 | | |

DOI: https://doi.org/10.7554/eLife.32548.010

course of the three rest periods in the Relearning phase to examine how they evolved over the progression of relearning and seek evidence of learning dynamics.

A property of the learning model is that PEs during the Relearning period should decrease from the period preceding Rest 1 (when participants are first encountering the new rewards in the revaluation condition) to the period preceding Rest 3 (after participants have had more time to learn these new rewards); this was the case for the PE values estimated by the learning model (*Figure 5*, top left). Given the decreasing pattern in unsigned PE during re-learning, we predicted that this within-block variation might be echoed in patterns of replay across rest periods – reduced PE should lead to less replay. To focus the analysis on differences among rest periods within each block, we mean-centered MVPA evidence for the Stage I stimulus during the three rest periods for each block of each subject. This allowed us to compare patterns of within-block differences between replay in the three rest periods of revaluation and control trials. In control blocks, we found no differences in MVPA evidence for S1 replay during the three rest periods (*Figure 5*). During revaluation blocks, however, evidence for offline replay significantly decreased from Rest one to Rest 3 ($t(23) = 2.9$, $p = 0.007$), similar to the decreasing pattern of unsigned prediction errors (*Figure 5*). Furthermore, offline replay was higher during Rest 1 of revaluation blocks compared to the Rest 1 of control blocks ($t(23) = 2.17$, $p = 0.039$). Thus, consistent with our hypothesis, prediction errors and replay track each other over the course of the block, as well as across blocks (as previously shown in *Figure 3*).

Analyzing the rest periods of revaluation trials in the noisy-rewards and noiseless-rewards conditions separately (*Figure 5—figure supplement 1*), we found a similar trend: In the noisy-rewards condition, replay was marginally higher during Rest one than both Rest 2 ($t(23) = 2.07$, $p = 0.049$) and Rest 3 ($t(23) = 2.02$, $p = 0.05$) – in the noiseless-rewards condition, replay levels were low overall and did not vary across rest periods.

Finally, we considered how this replay related to subsequent replanning behavior in the Test phase. We compared the replay-replanning correlation across the different rest periods (*Figure 5*; unpacking over time the correlation from *Figure 2B*). We found that, for the average of all revaluation blocks, this correlation was numerically stronger (and only statistically significant) during later rest periods (however, there was no significant difference between the correlation coefficient for Rest one versus the correlation coefficient for Rest 3, Fisher's $z = −1.01$, $p = 0.31$). We then analyzed this effect separately for revaluation blocks in the noisy rewards and revaluation blocks in the noiseless-rewards condition (*Figure 5—figure supplement 1*), and found again that the replay-behavior correlation was mainly driven by the noise condition, where the most significant replay-behavior correlation is observed during Rest 2.

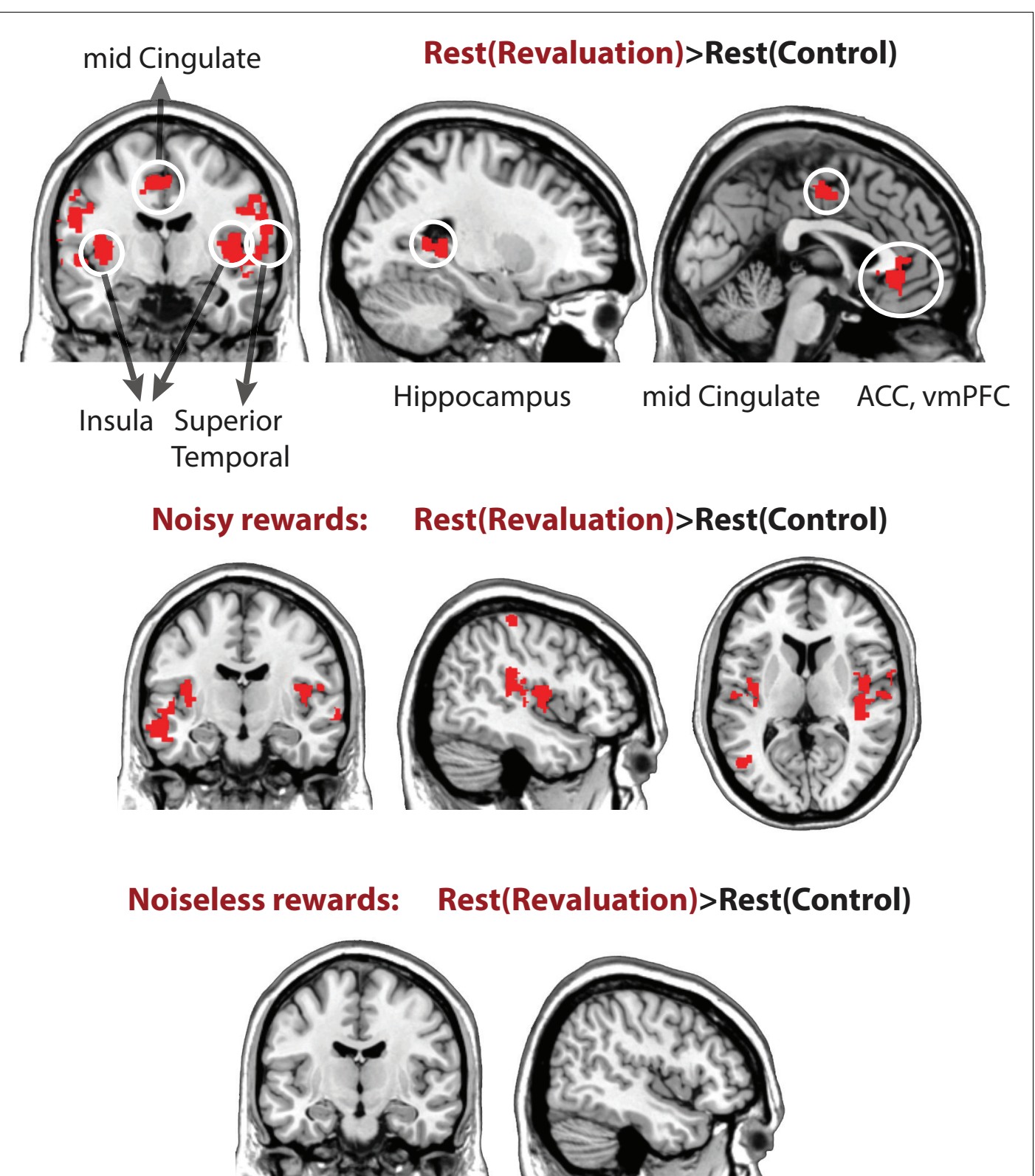

**Figure 4.** Univariate general linear model contrasts comparing activity rest periods of revaluation vs. control blocks, Rest$_{revaluation}$> Rest$_{control}$. The contrast reveals higher activity in the hippocampus, the anterior cingulate cortex, mid cingulate (shown above), as well as bilateral insula and superior temporal cortices (extent threshold p < 0.005, cluster level family-wise error corrected at p < 0.05). See *Table 3* for coordinates, cluster size, and p values.

*Figure 4 continued on next page*

*Figure 4 continued*

DOI: https://doi.org/10.7554/eLife.32548.012

The following figure supplement is available for figure 4:

**Figure supplement 1.** Differences in off-task univariate activation in revaluation vs. control.

DOI: https://doi.org/10.7554/eLife.32548.013

## Discussion

We present fMRI evidence for the hypotheses that uncertainty-driven replay can help integrate past episodes offline, and in turn, update representations and policies that support planning (*Van Seijen and Sutton, 2015*). We address two questions. Our main question was: does off-task replay support planning? Inspired by the Dyna architecture in reinforcement learning, our main contribution is to demonstrate a clear and selective association between offline replay and change in choice preferences, that is replanning. We found that MVPA evidence for offline replay during rest correlates with subsequent choice behavior in reward revaluation. We suggest that during offline replay the brain integrates and pieces together simulated trajectories of information acquired during multiple learning episodes and updates planning policies accordingly.

Our secondary question was: what predicts such offline replay? Inspired by prioritized sweeping in reinforcement learning, our second hypothesis was that certain classes of events – such as surprising ones – privilege trajectories for future replay that helps the brain to learn whether they have consequences for choice preferences. We examined this question using a representative index of surprise, the brain's response to unsigned prediction error. Whether this subsequent replay leads to revaluation or not will depend on whether the replayed experiences themselves support a change in behavior. Therefore, not all prediction errors will lead to revaluation, even if they increase replay. We found that the brain's sensitivity to unsigned prediction errors in the ACC, basal ganglia, and posterior medial cortex (the precuneus) across subjects and blocks predicted both the extent of subsequent offline replay as well as subsequent replanning behavior. We suggest that the brain's sensitivity to surprising information and increased uncertainty, here signaled by unsigned prediction errors, may tag certain episodes and trajectories of experience for selective offline replay later on (*Moore and Atkeson, 1993*; *Peng and Williams, 1993*).

Finally, we observed a general increase in hippocampal, posterior medial, anterior cingulate (extending to ventromedial prefrontal cortex upon relaxing the threshold), and superior temporal cortex activity during the rest periods of revaluation trials compared to control. The hippocampus has long been identified as the locus of episodic memory (*Ranganath and Ritchey, 2012*), posterior medial cortex is shown to be involved in recall and integration of memories over long time-scales (*Chen et al., 2017*; *Shenhav et al., 2013*), the anterior cingulate cortex is known to enable comparisons and signal conflicts the resolution of which may be valuable enough to recruit controlled processes (*van Kesteren et al., 2013*), and the ventromedial prefrontal cortex has been established as a region involved in memory retrieval of relevant information (*Preston and Eichenbaum, 2013*; *Gläscher et al., 2009*), value representation (*Fellows and Farah, 2007*), and decision making

**Table 3.** .Coordinates of peak voxels of regions with higher off-task activity during rest periods of revaluation > control condition. The clusters were selected with threshold p < 0.005, corrected at cluster level family-wise error p < 0.05 (these correspond to red regions in *Figure 4*).

| Region (cluster) | Z score | X | Y | Z | K voxels | P |
|---|---|---|---|---|---|---|
| Left hippocampus | 4.74 | −20 | −38 | 10 | 297 | .014 |
| Left insula | 4.57 | −36 | -8 | 6 | 1959 | .000 |
| Left superior temporal | 4.57 | −42 | −24 | 6 | | |
| Right superior temporal | 4.11 | 44 | −12 | 2 | 2249 | .000 |
| Right anterior cingulate, ventromedial PFC | 4.65 | 2 | 38 | 6 | 238 | .048 |
| Left mid cingulate | 3.31 | −10 | −10 | 46 | 258 | .030 |
| Left Supplementary motor area | 3.33 | -6 | −12 | 62 | | |

DOI: https://doi.org/10.7554/eLife.32548.011

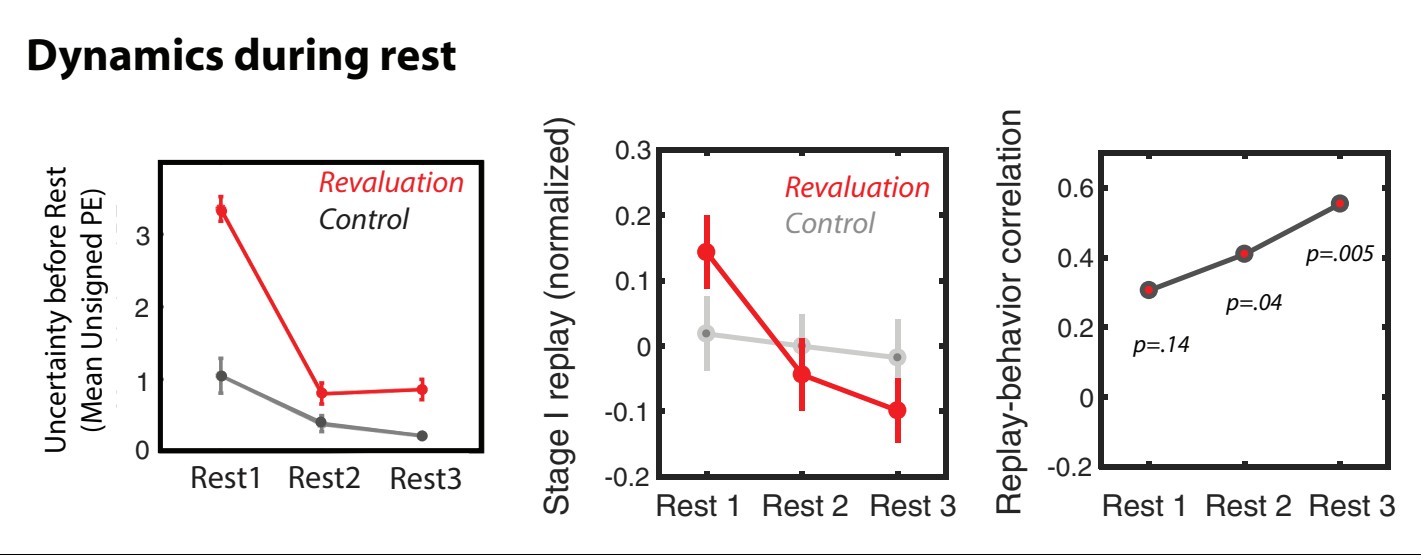

**Figure 5.** Dynamics of replay and activity during rest periods. Dynamics of prediction errors prior to each rest period (left), MVPA evidence for Stage I replay during each rest period (middle), and replay-behavior correlation across the three rest periods of revaluation runs (right).

DOI: https://doi.org/10.7554/eLife.32548.014

The following figure supplement is available for figure 5:

**Figure supplement 1.** Breakdown of replay during the three rest periods in the revaluation vs. control conditions during noiseless rewards and noisy rewards conditions (top) and the correlation of these replay magnitudes with revaluation magnitude (bottom) in the noiseless vs. noisy rewards conditions.

DOI: https://doi.org/10.7554/eLife.32548.015

(*Anderson et al., 2016*) respectively. Hippocampal-prefrontal interactions have been shown to mediate control of memory in memory inhibition (*Kalisch et al., 2006*) and updating memory in extinction (*Behrens et al., 2007*). Thus, this finding supports an increased role for memory and evaluation mechanisms during the rest periods of revaluation blocks.

In short, inspired by work on the Dyna architecture in reinforcement learning (*Sutton, 1991*) and prioritized sweeping (*Moore and Atkeson, 1993*; *Peng and Williams, 1993*) in machine learning, as well as behavioral paradigms of reward revaluation (*Gershman et al., 2014*; *Momennejad et al., 2017*), these results are the first to show a functional role for offline replay in planning under uncertainty. The key novel aspect of our results is our finding that offline replay can lead to replanning by *integrating multiple, temporally distal learning episodes* (here, computing the implication of Stage II experiences on choice policy at Stage I). The predominant assumption, going back to Tolman, is that humans and animals solve tasks of this sort by online computation, e.g. forward planning when a choice is required during the Test phase (*Tolman, 1948*), or earlier during acquisition. Accordingly, previous human studies have shown benefits of *online* reinstatement of past episodes on decision-making in a sensory preconditioning paradigm (*Wimmer and Shohamy, 2012*), and that goal-directed behavior in sequential decision-making is accompanied by the online reinstatement of prospective states (*Doll et al., 2015*). Other studies have explored properties of offline reinstatement, showing that it favors past episodes associated with higher reward (*Gruber et al., 2016*) and supports upcoming learning of related material (*Cichy et al., 2012*). However, to our knowledge, no human or rodent study has tested the role of offline replay in piecing together distal episodes to support planning. Tolman famously designed revaluation tasks that require the animal to integrate information experienced at various points in time (*Tolman, 1948*). Thus, we used a simplified Tolman-style revaluation task to test the role of offline replay in integration of distant memories and address the crucial question of which memories the brain would preferentially replay to improve planning. While the rodent literature has shown in various paradigms that offline replay is associated with better task performance (*Johnson and Redish, 2005*; *Ólafsdóttir et al., 2015*; *Ambrose et al., 2016a*), none verifies that these improvements are specifically due to the offline integration of

memory. Our present findings further these previous efforts and elucidate the role of offline replay in goal-directed memory integration that supports planning.

It is important to consider alternative hypotheses and interpretations of our results. An important general point is that our results are correlational, and we of course can't directly draw causal conclusions from the associations we report. Our main result is the association between offline neural reinstatement and subsequent preference change, supporting our hypothesis of a role for the former in replanning. In general, alternative hypotheses in the literature for how replanning occurs tend to focus on online computation. However, evidence for the importance of online computation is not mutually exclusive with the mechanism discussed here, and it cannot itself explain the reported association between planning and *offline* replay. In order to rule out alternative interpretations, we included a number of controls in the design of our study. As a general matter, the factorial design of our experiment, which includes no-revaluation controls and noise manipulations, and our use of a relative replanning score to control for initial learning, all help to verify that the association we report between replay and behavior is specific to replanning. For instance, if the brain-behavior relationship was confounded by many sorts of nonspecific factors such as variation in overall attention or motivation, or by forgetting rather than strategic replanning, similar brain-behavior relationships should be present in the control blocks as well. Accordingly, we can rule out the alternative interpretation that our results are due to behavioral change driven by 'any' surprising event regardless of whether it warrants a change in behavior. To control for this possibility, we included a noisy control condition where participants experienced prediction errors in the experienced rewards but the global structure of the task, and therefore the most rewarding choice, did not change. We did not observe a relationship between replay and revaluation behavior in this control condition. A second alternative interpretation is that any replay, regardless of the category, is associated with revaluation of choice behavior. Such a relationship between nonspecific replay and behavior could reflect a more generic index of task engagement, e.g., higher attention or motivation in participants who are better at the task. This interpretation would predict a relationship between Stage two category replay and replanning as well. However, we found no significant correlation between Stage two replay and replanning (*Figure 2—figure supplement 2*). Altogether, the specificity of the findings is consistent with the hypothesis that learning from simulated experience during replay determines whether revaluation occurs.

Our second key result concerns a further pair of associations, between earlier signatures of PE and subsequent replay and behavioral planning. This is consistent with a general family of explanations, in which events with certain characteristics (e.g., surprising one) are privileged for later replay. We operationalize this in terms of the neural response to unsigned prediction error during encoding, as suggested by prioritized sweeping (the standard embodiment of this general idea in computer science), but this is meant to be representative of a broader class of heuristics. Our task design does not definitively allow us to distinguish this particular heuristic measure of surprise from other, correlated ones that could support similar effects. For instance, in more elaborate models, overall variability in outcomes can be decomposed into contributions of volatility and stochasticity, and PE can be scaled relative to these (*Ambrose et al., 2016b*). Similarly, there is evidence that neural replay is sensitive to the sign of prediction errors (*Gläscher et al., 2010*). However, in the present task, signed and unsigned PEs are highly correlated (in particular, most PEs are positive, because subjects are incentivized to make rewarding choices and not return to disappointing states). Thus, the present design does not have sufficient power to dissociate the role of signed vs. unsigned prediction errors in replay.

Furthermore, although prioritized sweeping works by marking high-PE states when they are experienced – and this echoes many notions of 'tagging' in the cognitive neuroscience of memory – given the correlational design we can't rule out other mechanisms by which the decision which events to replay happens later. For instance, in principle, even without tagging, the brain could retrieve memories offline in an initially unbiased way to search for high-PE ones to focus on. Our data do not directly dissociate the two hypotheses, and both are consistent with our core contribution: that offline replay of surprising events supports planning. Having said this, we noted that, in realistic scenarios (as well as task designs with larger decision trees) such unguided search would be computationally expensive and potentially intractable. We experience many events and form many memories on a daily basis. In order to enhance learning and planning, it would be useful for the brain to have a mechanism to focus simulated experience on those that may lead to policy updates.

Thus, while acknowledging other possibilities, we suggest that some form of prioritized replay is more likely.

Given that our interest is primarily in the relationships between prediction errors, replay, and revaluation behavior, we focused our analysis on the difference between revaluation and control conditions, collapsing over blocks with noisy vs. noiseless rewards. We did, however, also consider all results separately across those conditions. Perhaps unsurprisingly (given lower power), we did not find significant differences between the conditions in any of our measures, but we note a few findings. First, we only observed a significant correlation between replay and replanning behavior in revaluation blocks in the noisy reward condition (*Figure 2—figure supplement 1*). These results may be due to a ceiling effect in the noiseless reward conditions, where updating past policies was sufficiently easy for online processes to adequately do the job. Second, we only found significant conjunction results for the PE analyses in the combined condition, but neither in the noisy-rewards condition nor the noiseless-rewards conditions alone. This is most likely due to the fact that the data in the noisy and noiseless conditions were half the number of runs included in the combined analysis reported in the main text (*Figure 3*). Future experiments should focus on the differential effect of the noisiness vs. stability of reward structures, and the volatility of event and reward structures more broadly, on brain regions recruited in offline processes that support planning. One possibility is that the brain is more likely to benefit from offline replay for planning when the reward structure in the world is noisier, in which case changes in mean reward may be neglected as variance, making online integration more difficult. While the present study has focused on reward prediction errors, future studies are required to study the role of other forms of prediction errors, e.g. state prediction errors (*Dunsmoor and Murphy, 2015*).

We also examined the dynamics of our results over the progression of the relearning phase. The key difference between the three rest periods was in the amount of unsigned prediction error (PE, or uncertainty) experienced prior to each (*Figure 5*). Here, unsigned prediction errors are how the brain notices something is not right, i.e. some prediction does not match the observation, and needs to determine whether the surprising experience may have further consequences for the brain's stored representations and policies (its model of the world). Before Rest one during revaluation trials, participants had just noticed that rewards have changed from what they expected, leading to the experience of high unsigned PEs while making Stage II decisions. Importantly, they did not visit the Stage I state during the Relearning phase, but only Stage II stimuli, therefore they did not experience any PEs after choosing a policy from Stage I. Before Rest 3 of revaluation trials, participants experienced smaller prediction errors because they had already had the chance to learn the new reward structures for Stage II stimuli; correspondingly, overall levels of replay were lower after Rest three than after Rest 1. However, crucially, while levels of replay were lower after Rest 3, there was still a robust correlation between Rest three replay and behavior (the size of the replay-behavior correlation did not significantly differ across rest periods).

Taken together, these findings suggest a potential two-stage functional role for replay in the present study. First, replay of trajectories marked with unsigned PE could allow identifying distal parts of the state space that may need to be updated. When the brain encounters prediction errors, initial replay can enable search through a graph of memory states to identify other states affected by the experienced PE: this is the 'sample to tag with priority' function of replay. The present experiment consists of one such past state, but it is possible that in larger decision trees this search for relevant past states leads to 'reverse replay' (*Wu and Foster, 2014*; *Ambrose et al., 2016a*). As such key past states that lead to the present state in which high PE was experienced may be tagged with priority over other memories to be replayed later on. Second, once the new structure of the environment is adequately learned, and high PEs are no longer experienced; the tagged relevant past states and their corresponding trajectories can be replayed, simulated if never directly experienced, and updated. A similar tagging idea had been proposed in the domain of fear learning, where memory traces could be retroactively altered in accordance with new behavioral relevance of fear associations (*Kurth-Nelson et al., 2016*), whereas here we propose a more general case where unsigned PEs tag memories with priority for subsequent integrative replay. As such, the second function of replay may be 'learning from replay': to actually enable the brain to piece together and simulate trajectories in order to update past representations and planning policies offline (*Pfeiffer and Foster, 2013*).

It is worth commenting on how the notion of replay used here relates to the notion of sequential replay in the rodent literature – in this literature, the term 'replay' often refers to the replay of sequences at sub-second speed. The temporal resolution of fMRI poses a challenge to providing direct neural evidence of sequential replay. That said, we designed the study such that replanning the optimal Stage I choice required integration of past information by piecing together different memories sequentially: The transition from Stage I to Stage II, on the one hand, and the new Stage II rewards, on the other, were originally experienced at separate distal points in time during online task performance. Unlike animal studies, but similar to previous human fMRI studies of offline replay (*Staresina et al., 2013*; *Cichy et al., 2012*; *Gruber et al., 2016*), here we leverage a controlled design together with evidence of Stage I reinstatement to investigate the role of offline replay in planning. Future work using methods with higher temporal resolution, such as MEG and direct recordings from the hippocampus in patients, is required to shed light on the details of forward and reverse sequential replay in updating representations and memories that serve planning.

In future work, we plan to conduct more detailed explorations of the functional role of replay in planning, using paradigms with larger decision trees and higher temporal resolutions that make it possible to track replay trajectories in more detail; e.g., a recent human study used MEG to identify a role for reverse replay of trajectories (*Dayan, 1993*). We also plan to explore how offline replay might contribute to generalization and temporal abstraction, e.g. via multi-step predictive representations of upcoming states (or the successor representation, (*Russek et al., 2017*; *Momennejad et al., 2017*). Caching of multi-step predictive representations during replay can in turn enable generalization (*Botvinick and Weinstein, 2014*), faster planning especially under time pressure, and sub-goal discovery (*Beckmann et al., 2003*) across longer scales. After consolidation, the generalized multi-step representations could lead to abstract representations of states and policies such as schemas. Using generalized representations, both discovery of relevant past states (as in the first function of replay above) and piecing together episodes to simulate trajectories to goal (as the second function of replay above) could also take place in a multi-step fashion consistent with the successor representation (*Russek et al., 2017*), enabling more efficiency simulation of past and future trajectories.

To summarize, we have provided neural and behavioral evidence for the hypothesis that offline replay supports planning. We further suggest that the brain's sensitivity to uncertainty, operationalized as unsigned prediction errors, mediates the amount of offline replay and, through this, revaluation behavior. Our findings further our understanding of how the brain leverages offline memory mechanisms in planning and goal-directed behavior.

## Materials and methods

### Subjects

We recruited 26 volunteers using the Princeton University recruitment system (SONA) to participate in the fMRI study. Two participants were excluded due to running out of time and not having sufficient number of runs. Please note that, logically, the analysis required each participant to learn the local dependencies of the task in the learning phase (indexed by their choice behavior) and there was one run per condition. If a participant did not learn the task structure during the learning phase, neither their replay during rest nor their subsequent choice behavior could be interpretable. That is, for any 'replanning' to occur later on, participants needed to learn sufficiently to be able to show 'planning' behavior in the learning phase, indexed by their choices. Therefore, based on this a priori requirement, any participant who did not learn the task structure during the learning phase was logically excluded. The remaining 24 participants' data were further analyzed. The Princeton University Institutional Review Board approved the study (Protocol #6014, 'Cognitive Control: Functional Imaging of Memory and Mental Representations'). All participants gave informed to participate in the fMRI study and signed a screening form that ensured they had normal or corrected to normal vision, had no metal in their body, and had no history of psychiatric or neurological disorders.

### Stimuli and task

Eight face, eight scene, and eight object stimuli were used in the face/scene/object functional localizers. The faces and scenes were subsequently used in the main experimental task. Before the

beginning of the main experiment, participants were subjected to blocked functional localizer runs. In the localizer participants were exposed to separate blocks of scenes, faces, and objects. Each category was presented for four blocks that each lasted for 12 s, separated by rest periods of 12 s. In each block, every stimulus was presented for 3.5 s with an ISI of. 5 s. To ensure that participants were paying attention to the stimuli during the localizer, they were instructed to give responses to a cover task. In half of the localizer blocks participants were instructed to give responses with the left hand and in the other half of the blocks with the right hand.

The main experimental paradigm consisted of 4 runs, each of which corresponded to one Markov Decision Process (MDP) with three phases. Two of 4 runs were assigned to the revaluation condition and the rest were control runs (in which no revaluation took place). The order of conditions was randomized for each participant. The task was explained to the participants in terms of a cover task with a robbery scenario: They were told the experiment was a game where they were to explore different locations in different cities, each corresponding to each run's simple 2-step maze to find out which state they could steal more money from. Participants were instructed that they would receive a bonus compensation for their performance on the final Test phase, which they received for all runs at the end of the fMRI session. Each MDP was presented as a new 'city' in which the participants would explore a building in search of money. Different cities were randomly assigned to different conditions across participants. Each MDP consisted of 3 main states: Stage I and two Stage II states. The same stimulus signaled Stage I in each city, and the same Stage II stimuli signaled states 2 and 3. Importantly, Stage I and Stage II stimuli belonged to different categories. In 2 out of 4 cities state one was a face and states 2 and 3 were scenes, and in the remaining two cities the opposite was the case.

Additionally, we manipulated the variance in the rewards observed at each state. We wanted to ensure that, after learning, participants did not only experience prediction errors in the revaluation condition, but that the control condition also contained some degree of uncertainty with respect to rewards. To this end, the rewards in half of the runs, in other words one revaluation run and one control run, were sampled from a normal distribution with a fixed mean and variance, and in the other half of runs the rewards were completely deterministic.

In each city, during the Learning phase participants were exposed to the stimuli associated with the three main states of the MDP about 70 times. The last 10 choices they made from the Stage I state were used to calculate their probability of choosing the optimal response (e.g. if the optimal policy was left: p(left) = #left choices/10). Following the Learning phase, participants entered the Relearning phase where they only visited Stage II stimuli and never the Stage I stimulus. During this phase in revaluation trials, the rewards associated with Stage II (state 2 and 3) choices changed – but not during control trials. After every six Stage II episodes each requiring a choice, a rest period of 30 s length followed for which no instructions were given. Each participant experienced three rest periods of 30 s each during the Relearning phase. Finally, during the Test phase, participants were once again exposed to state 1, four times. Replanning magnitudes were computed as the probability of choosing the other valid action in those four trials, minus the probability of choosing the same response during the Learning phase – thus detecting the actual change in performance by subtracting noisiness in decision at the end of the Learning phase. For instance, if the optimal policy at test was right: revaluation magnitude = (#right choices at test/4) - (#right during last 10 choices of learning/10).

$$replanning\ magnitude = \frac{ns_{test}}{n_{test}} - \frac{ns_{learning}}{n_{learning}} \qquad (1)$$

Here $n_{test}$ denotes the total number of valid responses (i.e., excluding missed responses or invalid button presses) during the Test phase, and $ns_{test}$ is the number of times that the optimal choice was made during the test phase; $n_{learning}$ denotes the total number of valid responses during the last 10 trials of the Learning phase, and $ns_{learning}$ is the number of times that the choice that was optimal at Test was selected during the last 10 trials of the learning phase (note that, in revaluation blocks, the response that is optimal during the Test phase was suboptimal during the Learning phase).

## fMRI imaging protocol and preprocessing

Imaging data were acquired on a 3 Tesla Skyra scanner (Siemens, Erlangen, Germany) using a 20-channel head coil. Functional runs were acquired using T2*-weighted echo-planar sequence with 37 slices (interleaved order, ascending) with 3 mm thickness, an in-plane resolution of 3 × 3 m, TR = 2.08 s, echo time of 70 ms, anterior-to-posterior phase encoding direction, and no gaps. Slices were tilted for each participant by roughly 30 degrees to avoid the sinuses and optimize signal in the orbitofrontal cortex but avoid losing temporal cortex voxels. A T1-weighted structural volume was acquired with 1 × 1×1 mm resolution. Functional volumes were slice-time corrected, realigned to the mean of each run, and motion corrected. FMRI data processing was carried out using FEAT (FMRI Expert Analysis Tool) Version 6.00, part of FSL (FMRIB's Software Library, www.fmrib.ox.ac.uk/fsl). Higher-level analysis was carried out using FLAME (FMRIB's Local Analysis of Mixed Effects) stage 1 (*Woolrich et al., 2004*; *Woolrich, 2008*; *Kuhl et al., 2011*).

## Behavioral and neuroimaging data analysis

**Functional localization** We designed a general linear model (GLM) to identify functional regions for each participant. We used regressors corresponding to face, scene, objects, and rest conditions, and applied Scene > Face and Face > Scene contrasts to identify face-selective and scene-selective regions for each participant (*Eichenbaum, 2004*); for these contrasts, we used a liberal voxel-wise threshold of $p < 0.005$, uncorrected, to ensure that we did not miss informative voxels. These regions of interest were used as a means of dimensionality reduction, allowing us to confine the multivariate analysis of experimental runs to these regions.

## Multivariate pattern analysis

As noted, Stage I stimuli were of a different category than Stage II stimuli. For each participant, Stage I stimuli were faces in half of the blocks and scenes in the other half. This ensured that identifying Stage I evidence was not confounded with simply a replay bias for faces and scenes, which might have been the case if the Stage I stimuli were from the same category across all trials. We used L2-regularized logistic regression (penalty = 1) to train a classifier on TRs when participants observed images of faces, objects, and scenes during the independent functional localizer run acquired prior to experiment (*Polyn et al., 2005*). Importantly, to detect face activation, we trained a face vs. other (scene, object) classifier only on voxels from the face-selective ROI identified in the localizer analysis above; conversely, to detect scene activation, we trained a scene vs. other (face, object) classifier only on voxels from the scene-selective ROI identified in the localizer analysis above. Our use of this approach (where we used separate sets of voxels for face detection and scene detection) was driven by our desire to obtain, to the greatest extent possible, *independent* readouts of face and scene activity. It is a well known issue with MVPA that, when face and scene classifiers are applied to the same voxels, face and scene classifier readouts tend to be artifactually anti-correlated because the face and scene labels are anti-correlated in the training data (i.e., the classifier learns that, in the training data, the presence of scene predicts the absence of face, and vice-versa) (*Wilson and Niv, 2015*). This problem can be mitigated to some degree by adding additional categories at training (this is why we added the object category), but we found that it can be mitigated even further by applying the classifiers to distinct sets of voxels – this carries a potential cost of losing sensitivity (by restricting the voxels going into a particular classifier) but we were willing to pay this price to address the anti-correlation problem described above: In our study, the correlation between the time course of face evidence (from face ROIs) and scene evidence (from scene ROIs) during the rest periods was *positive* (on average,. 36), not negative.

To measure replay of Stage I category information, we took the trained face and scene classifiers and applied them to the TR-by-TR volumes from each run's rest periods, focusing specifically on the last 10 TRs of each rest period (we omitted the first 5 TRs from each rest period to minimize 'spill-over' from participants perceiving Stage II stimuli right before the rest period started). As noted above, if the Stage I stimulus was a face, we used the face classifier to detect replay; if the Stage I stimulus was a scene, we used the scene classifier. For each TR, the classifier outputs an evidence value for the selected category between zero and one; we averaged these evidence values across TRs to get a neural replay score for each block (since there were three rest periods per block and we used 10 TRs per rest period, the replay score for each block was based on 30 TRs).

The main goal of this analysis was to test whether evidence for offline replay of the root state (State 1) during the rest periods correlated with subsequent revaluation behavior during the Test phase. Therefore, we computed correlations between mean classification accuracy (evidence for offline replay of Stage I) and subsequent revaluation behavior separately for revaluation condition trials and control trials (*Figure 2B*).

## Bootstrap comparison of replay-replanning correlation across conditions

To compare the size of the correlation between replay and behavior across conditions (revaluation vs. control), we used a bootstrap approach in which we resampled participants with replacement 1000 times; for each bootstrap sample, we computed the difference in the size of the replay-behavior correlation across conditions, thereby giving us a bootstrap distribution of correlation differences.

## Univariate analysis of rest periods

### Model-based analysis

In order to identify regions that were sensitive to reward prediction errors (RPE), we ran a parametric modulation analysis in which we regressed out mean signal change when participants received reward, and identified regions where variance around the mean was modulated by the magnitude of absolute RPEs experienced while receiving the reward. Unsigned prediction errors were calculated for each subject and each stimulus using *Equation (2)*.

$$|\delta| = |\mathrm{r}\left(s^{'}\right) + \gamma Q\left(s^{'}, a^{'}\right) - Q(s, a)| \tag{2}$$

Here *s* refers to the current state and *s'* to the next state, *a* is the current action and *a'* the next action, and $\gamma$ is the temporal discounting parameter (here $\gamma = 1$, since the temporal horizon in a 2-step Markov decision process (MDP) is limited to one step (*42*)). The reward prediction error is in turn used to update the state-action value as equation (3) below, where $\alpha$ denotes the learning rate. We estimated a learning rate of. 7 based on analysis of previous pilot behavioral data (n=24) and used this learning rate for all participants.

$$Q(s, a) = Q(s, a) + \alpha \delta \tag{3}$$

We were interested in identifying regions where sensitivity to unsigned reward PEs, a measure of uncertainty used in the reinforcement learning literature, would correlate with future replay during rest periods and revaluation magnitude during the final Test phase.

## Estimating learning rates

We estimated learning rates from a separate sample that performed the same task in a pilot study. The structure and rewards used in this pilot study were virtually identical to the fMRI task. As model-based fMRI analyses are typically performed using group-level (rather than individual level) parameter estimates, we estimated this learning rate from the pooled subject data using Maximum Likelihood Estimation, resulting in one best-fitting learning rate for the entire population (learning rate of. 7, *n = 24*). We confirmed a similar best-fitting learning rate in the fMRI sample (.78). With regard to the generalizability of the neural results to different values of the learning rate, it is worth noting that a previous systematic evaluation of this issue (*51*) found that substantive differences in learning rates in the model-based analysis of similar tasks lead to only minute changes in neural results.

## The effect of volatility on prediction errors

Here we have estimated prediction errors regardless of the noise conditions. However, theoretically the statistical volatility of, and hence the overall uncertainty about, the environment or context may change the extent to which prediction errors tag memories for offline replay. For instance in environments with high volatility or uncertainty, the brain may become less sensitive to PEs that are not larger than some threshold. One possibility to give a computational account of the effect of such contextual uncertainty (or reward volatility) is to divide the PE by the variance (e.g. in rewards) in future models. Such a simple computational step, and similar varieties using the hazard rate, would

lead to lower prediction errors in contexts with higher variance. Our data only partially speaks to this point: the noisy revaluation condition (that includes uncertainty about the precise reward outcome) leads to worse replanning performance than the noiseless revaluation condition (that has no uncertainty about outcomes outside of the revaluation). A systematic investigation of the role of hazard rate or volatility of rewards on tagging of surprising memories for future replay merits a separate dedicated study. Such a study would benefit from a design where prediction errors are parametrically changed across conditions or groups.

## Acknowledgments

We thank Matthew Botvinick and Sam Gershman for helpful conversations, and Sarah Dubrow, Dylan Rich, and James Antony for helpful comments on manuscript drafts. We acknowledge and thank Eeh Pyoung Rhee, who ran control analyses to compare classifier performance for his senior thesis in computer science at Princeton University. This project and publication was made possible through the support of a grant from the John Templeton Foundation, grant 57876, and NIMH grant R01MH109177, part of the CRCNS program. The opinions expressed in this publication are those of the authors and do not necessarily reflect the views of the John Templeton Foundation or other funding bodies.

## Additional information

### Funding

| Funder | Grant reference number | Author |
| --- | --- | --- |
| John Templeton Foundation | 57876 | Ida Momennejad<br>Kenneth A Norman |
| National Institute of Mental Health | R01MH109177 | Nathaniel D Daw |

The funders had no role in study design, data collection and interpretation, or the decision to submit the work for publication.

### Author contributions

Ida Momennejad, Conceptualization, Data curation, Software, Formal analysis, Supervision, Validation, Investigation, Visualization, Methodology, Writing—original draft, Project administration, Writing—review and editing; A Ross Otto, Conceptualization, Data curation, Formal analysis, Methodology, Writing—review and editing; Nathaniel D Daw, Kenneth A Norman, Conceptualization, Resources, Software, Supervision, Funding acquisition, Methodology, Project administration, Writing—review and editing

### Author ORCIDs

Ida Momennejad (iD) http://orcid.org/0000-0003-0830-3973
A Ross Otto (iD) http://orcid.org/0000-0002-9997-1901
Nathaniel D Daw (iD) http://orcid.org/0000-0001-5029-1430
Kenneth A Norman (iD) http://orcid.org/0000-0002-5887-9682

### Ethics

Human subjects: The Princeton University Institutional Review Board approved the study. All participants gave informed consent to participate in the fMRI study and signed a screening form that ensured they had normal or corrected to normal vision, had no metal in their body, and had no history of psychiatric or neurological disorders.(Protocol#6014).

### Decision letter and Author response

Decision letter https://doi.org/10.7554/eLife.32548.024
Author response https://doi.org/10.7554/eLife.32548.025

# Additional files

## Supplementary files
• Transparent reporting form
DOI: https://doi.org/10.7554/eLife.32548.016

## Data availability
Neural and behavioral data have been made available online at OpenNeuro (https://openneuro.org/datasets/ds001612/versions/1.0.0).

The following dataset was generated:

| Author(s) | Year | Dataset title | Dataset URL | Database and Identifier |
|---|---|---|---|---|
| Ida Momennejad, A Ross Otto, Nathaniel D Daw, Kenneth A Norman | 2018 | Neural and behavioral data from Offline Replay Supports Planning in Human Reinforcement Learning | https://openneuro.org/datasets/ds001612/versions/1.0.0 | OpenNeuro, 10.18112/openneuro.ds001612.v1.0.1 |

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
