## [Decision Letter]

Thank you for submitting your article "Offline Replay Supports Planning: fMRI Evidence from Reward Revaluation" for consideration by *eLife*. Your article has been reviewed by two peer reviewers, and the evaluation has been overseen by a Reviewing Editor and Michael Frank as the Senior Editor. The following individual involved in review of your submission has agreed to reveal his identity: Fiery Cushman (Reviewer #4).

The reviewers have discussed the reviews with one another and the Reviewing Editor has drafted this decision to help you prepare a revised submission.

Summary:

This manuscript investigates how people engage in offline replay to update their value estimates decisions during a sequential task. Specifically, the experiment constructs a situation in which participants assign value to early decisions based on late outcomes, and then acquire new information about late consequences outside the context of early decisions. This provides a putative opportunity to compute the new expected value of the early decisions through offline replay. The paper reports behavioral and MVPA evidence that offline replay aids in this type of re-planning.

Essential revisions:

The editors and reviewers agreed that this paper investigates a fundamental topic. Further, the approach that combines MVPA estimates of task state with a two-step task is innovative and clever. However, both reviewers also raised significant concerns regarding the design and analysis. The most serious of these questioned what conclusions can be drawn from these results and whether there is specific evidence for planning versus other alternatives. In discussion among the reviewers and editors, it was agreed that several essential revisions are critical to shore up the specific contribution here beyond a basic association of replay with performance.

1) Alternative accounts of these results.

The authors consider the possibility that unsigned reward prediction errors experienced during the "relearning" phase prompt revaluation. This is plausible, but the authors don't consider any alternative hypotheses. In the absence of an alternative, it's hard to evaluate whether the evidence they consider in favor of this model is actually decisive – i.e., does their evidence uniquely favor their model, or is it consistent with other models?

Moreover, they undercut their own model when they write: "the inclusion of reward noise in the noisy-rewards condition (even in control blocks) allowed us to rule out the possibility that replanning is brought about by 'any' experience of uncertainty, whether or not replanning was optimal". This is a good point and an elegant feature of their design, but they don't seem to address how their model of revaluation assesses the "optimality" of replanning beyond unsigned PEs. In other words, they include a control condition specifically to show that their model is incomplete, and then don't address how it could be made complete.

Just to sharpen this point, here is one alternative account of what prompts revaluation. Possibly, during the rest period, people start engaging in value iteration from random points in their task model, which includes Stage 1. They then continue the process of value iteration at all points where value iteration results in high unsigned PEs during update. In other words, when performing value iteration on Stage 1 actions, the mismatch between the existing Stage 1 value estimates and the newly updated Stage 2 value estimates generates high unsigned PEs. These remain high until sufficient value iteration occurs that Stage 1 actions are appropriately revalued.

In summary, perhaps what drives the total amount of revaluation is not the magnitude of Stage 2 PEs during online relearning, but rather the magnitude (or persistence) of Stage 1 PEs during offline revaluation. (It should be clear how this stands in contrast to the "tagging" idea presented in the general Discussion. Incidentally, there is no reason that both things couldn't happen).

This alternative model addresses the question of why there is no reevaluation for high variance rewards in the control condition: the variance of the terminal rewards does not affect the Stage 1 PEs during value iteration.

It also seems to be consistent with all the data the authors present, because (except in the control condition) high unsigned Stage 2 PEs during relearning will be correlated with high unsigned Stage 1 PEs during revaluation.

But this correlation need not always hold – the difference between the two models can be tested. For instance, imagine that during the revaluation phase, values are swapped on the right/left dimension within each state. i.e., if formerly State 2/left was $1 and State 2/right was $8, then now State 2/left is $8 and State 2/right is $1. On the authors' model, this will result in high unsigned PEs during relearning and therefore also high levels of revaluation. On the model sketched above, however, it would not result in high levels of revaluation because max(Q) for each Stage 2 state does not change, and so there is no change to the Q value of each Stage 1 state, and thus no PEs during value iteration.

(A conceptually similar experiment would involve drifting rewards away from their original values but then right back to their original values all within a single relearning phase. This results in high PEs during the phase, but the model sketched above would not predict much revaluation for reasons similar to those described above).

The purpose of detailing this alternative is not to argue that this model is right or the only alternative, but to illustrate a much broader point. By sketching only one model of what prompts revaluation and failing to explicitly consider alternatives, the authors leave the reader in a poor position to evaluate whether the analyses that are consistent with their model are in fact good evidence for it – we don't know whether those analyses are also consistent with all plausible alternatives. Could the authors show us what models of revaluation-prompting are ruled out by their analyses? This would be a big help, whether or not the authors decide to discuss the specific alternative mentioned here.

2) Alternatives raised by the design and logic

Akin to the above point, there were several points regarding the design that raise other alternatives, including basic issues of engagement. Specifically:

- What incentive did people have to go to the higher-valued states in the learning phase? It seems this part of the task was not incentivized, so people had no particular reason to select higher numbers over lower numbers (except for learning for later). The replanning magnitude metric then becomes potentially problematic in that it subtracts away performance during the learning phase, which was not incentivized. This issue becomes especially problematic when trying to assess the impact of the brain replay (e.g., Figure 2). Here replay correlates with replanning, but which aspect of the replanning is not clear. Perhaps this correlation is due to behaviour in the learning phases-i.e., those people who paid more attention and learned more initially show more replay later on. That's not so surprising or interesting. The interesting thing is to predict later behaviour, but subtracting away a baseline that precedes the replay means that the replay can just be correlating with something that occurred earlier (learning/attention/task engagement, not replanning). It would be more convincing to show that the replay correlates solely with the test behaviour not with this replanning metric.

- The above point is compounded somewhat with the choice of the control condition here. Unlike the revaluation condition, there is no learning necessary at all in the second phase. There is also no incentive to pay attention or be engaged at all during this second phase. As a result, the differences in brain activity may be due to differential engagement, rather than specific to the replay-replanning connection as claimed here.

- The test phase gave no feedback or guidance. How were participants supposed to know which experience to draw upon when making those decisions? The assumption here is that the optimal thing here is to integrate the information in the two phases by adjusting the values of the terminal states in phases 2 and keeping the same phase 1 knowledge of task structure. A smart participant, however, might infer that the recurrence of the Stage 1 stimulus (which never appeared in the relearning phase) meant that the Learning phase reward-contingency rules were back in effect. It's not obvious what is optimal here. Maybe they have learned that the Stage 2 stimuli give different values when they are (or are not) preceded by the Stage 1 stimuli (i.e., a type of occasion setting).

- What were the means/variances of the experienced distributions? Were the actual means of the experienced distributions controlled or only the generative process? Were the experienced distributions close to the planned ones? How was it ensured that all conditions were experienced? Could people have avoided one of the outcomes e.g., due to getting initially unlucky (hot-stove effect)?

- It was not clear is why non-specific replaying the Stage 1 stimuli during the replanning task would help. In some ways, it says that people who are more engaged with the task do better on the task. That's interesting, but not groundbreaking. The data here do not seem to show that category-specific replay predicts later performance. Does replay of other categories (not the Stage 1 stimulus) not correlate with replanning behaviour? That would be a good additional piece of evidence beyond simple task engagement.

3) Concerns over the robustness of the analysis and approach to the computational modeling. It should be noted in light of the first comment below that *eLife* does not require that all hypothesis, predictions, or analyses be preregistered. However, the reviewers agreed that there should be explicit discussion of investigator degrees of freedom and what safeguards against flexibility in the analysis were put in place. Further, any checks that can be provided for robustness would be helpful. Here are the specific comments raised regarding the analysis and modeling.

- The article is pitched as confirmatory in the sense that it makes predictions and then confirms those predictions with evidence. Were any of these predictions pre-registered anywhere? Were there any competing hypotheses? What sort of data might have gone against these predictions? The evidence in the paper could use to be bolstered to establish that it was truly a confirmatory study. Similarly, were any of the metrics or analyses preregistered? If they were indeed hypothesis-driven, that should have been possible. It would seem that there are many, many ways that these data could have been analyzed (both behavioural and neural), and some of the metrics seem very arbitrary (e.g., number of trials in learning phase). Were other analysis pathways attempted? Evaluating the import of the inferential statistics requires more knowledge of the researcher degrees of freedom.

- Two participants were excluded post-hoc due to "not having sufficient number of runs". Were there any pre-defined criteria on this front? Or was this exclusion made after the data were examined? Additionally, the sample size seems small (though not inordinately so for an fMRI study), leaving many of the statistical tests underpowered (and indeed many of the brain-behaviour correlations fell in the marginal zone, making interpretation difficult).

- The idea that high prediction error should be preferentially replayed seems like a rule that would only work in very narrow (static) environments. If there is any variance in the rewards, then this rule could lead to perverse replay of unreducible uncertainty.

- The selection of learning rate requires further justification. How was the.7 estimated? What sort of task? If the task were as simple as the current task, with such a wide gap in rewards, I wonder how reliable an estimate this number represents. Moreover,.7 seems extremely high for a task with variable rewards. That would make people extremely influenced by the most recent outcome, much more so than a typical task with variable or probabilistic rewards. Is it reasonable to use the learning rate estimated in a different sample for analyzing the new sample? How robust are the prediction error analysis to this selection of learning rate?

---

## [Author Response]

Essential revisions:The editors and reviewers agreed that this paper investigates a fundamental topic. Further, the approach that combines MVPA estimates of task state with a two-step task is innovative and clever. However, both reviewers also raised significant concerns regarding the design and analysis. The most serious of these questioned what conclusions can be drawn from these results and whether there is specific evidence for planning versus other alternatives. In discussion among the reviewers and editors, it was agreed that several essential revisions are critical to shore up the specific contribution here beyond a basic association of replay with performance.

We thank the editor and reviewers for their constructive comments – these comments have enabled us to substantially improve the manuscript. We have edited the manuscript accordingly. Below we address the reviewer’s points and quote edited excerpts from the manuscript in our response.

We will first summarize what we see as the main contributions of our paper, including alternative hypotheses that we have ruled out. In the revised manuscript, we have sought to be much clearer about what can and cannot be concluded from our results.After this overall summary, we respond to the reviewers’ concerns in a point-by-point fashion.

1) What are the consequences of replay?

Inspired by the Dyna architecture in RL: Our main contribution is showing that replay is correlated with successful replanning in the revaluation condition.

Our core hypothesis was that, in situations where the reward structure of the world has changed, offline replay would drive learning and revaluation (replanning). This hypothesis was inspired by the Dyna architecture proposed by Rich Sutton in machine learning (R. S. Sutton, 1991, Dyna, an Integrated Architecture for Learning, Planning, and Reacting), in which offline replay of experience trains an actor/learner. We designed our experiments to find evidence for Dynalike algorithms in the brain. In support of our hypothesis, we found that fMRI-replay of the Stage 1 category during the relearning phase was correlated with later replanning behavior in the revaluation condition. This relationship between neural replay and behavior was specific to the revaluation condition, not the control condition.

The finding that replay was associated with behavioral change in the revaluation condition but not the control condition rules out the possibility that replay is always associated with behavioral changeand also a number of other potential confounds involving attention or forgetting. Rather, it appears that (as we hypothesized) replay is only associated with behavioral change in situations where (prior to rest) the reward structure of the world changed. We regard this specific relationship between replay of an antecedent state and subsequent behavioral change as the most important contribution of the present manuscript. In the revised manuscript, we discuss these alternative hypotheses (and why they are ruled out by the data) in a much more direct fashion.

2) What predicts replay?

Inspired by prioritized sweeping in RL: Our second hypothesis was that surprise during encoding drives later replay.

The second contribution of our paper is to explore what sorts of events are favored for the replay that drives behavioral change. We approach this question by looking for factors at encoding that predict subsequent replay. Our hypothesis about this was inspired by prioritized sweeping in machine learning, which uses how surprising an experience is to prioritize its later replay. Surprise is operationalized here as the unsigned magnitude of a prediction error. As predicted, we found that the brain’s sensitivity to unsigned prediction errors in the ACC, basal ganglia, and posterior medial cortex (the precuneus) during initial encoding was correlated with the extent of subsequent offline replay. We also found a relationship between neural sensitivity to prediction errors and subsequent *replanning* in the revaluation condition.

However, as with replay, it is crucial to emphasize that according to our hypothesis there will not *always* be a relationship between surprise and revaluation. Surprise will drive replay, but whether or not this replay drives behavioral change will depend – as noted above – on whether the world has changed. Again, the lack of behavioral effects in the control condition speaks to the specificity of the mechanisms we are observing, and against more nonspecific alternatives in which behavioral and neural measures might generally covary due, for instance, to subject differences in task attention or motivation. In the revised manuscript, we clarify our predictions on these points, and in the remainder of this letter we show how these points address various concerns raised by the reviewers.

We have now edited the text to reflect these issues more clearly:

“Our main question in this study is: does replay predict a change in choice behavior? […] This replay, in turn, enables the integration of separate memories in the service of updating optimal reward policy (action policies that maximize reward). We combine model-based fMRI, MVPA, and a sequential decisionmaking task to test our hypotheses.”

And we added the following:

“We thus explored what events favor replay, and ultimately successful revaluation. […] We tested this idea in terms of one representative notion of surprise – neural sensitivity to unsigned PEs (i.e., the absolute value of PEs), as in the prioritized sweeping algorithm (Moore and Atkeson, 1993).”

We have also edited the first two paragraphs of the Discussion to make sure that the above points are clear to readers:

“We present fMRI evidence for the hypotheses that uncertainty-driven replay can help integrate past episodes offline, and in turn, update representations and policies that support planning (Van Seijen and Sutton, 2015). […] We suggest that the brain’s sensitivity to surprising information and increased uncertainty, here signaled by unsigned prediction errors, may tag certain episodes and trajectories of experience for selective offline replay later on (Moore and Atkeson, 1993; Peng and Williams, 1993).”

Furthermore, it is important to note that there are alternative hypotheses that are *not* specifically ruled out by our data. In the current revision, we attempt to more clearly delineate these possibilities. The first and most important point is that of course in a correlational study, we cannot establish that the neural events we observe are causal for behavior. In particular, based on prioritized replay we hypothesize that a surprising event is tagged for offline replay, but the reviewers suggest an alternative mechanism in which the decision of which experiences to replay occurs later (during rest). However, because this decision is based on the same criterion we posit (surprise), it predicts the same neural correlations. Note that this alternative hypothesis shares with our account the core computational idea that surprise triggers replay; the only difference is whether it is surprise during *relearning* or surprise during *replay* that actually boosts (subsequent) replay. In the revised manuscript, we now discuss this alternative hypothesis, along with some other, computational reasons to favor our hypothesis over this alternative. See our response below, c.f.1) Alternative accounts of these results.

A second caveat, which we also now state clearly in the revised manuscript, is that it is possible that these results might reflect an effect on replay of some quantity other than absolute prediction error, but which is correlated with it. One possibility here is the *signed* prediction error (which would favor replay of better – but not worse-than-expected events). There are some computational (Mattar and Daw BioRxiv, 2017) and empirical (Ambrose et al., 2016) suggestions supporting this alternative hypothesis. Unfortunately, as we now state in the manuscript, our experiment is not designed to distinguish these quite similar alternatives. Please see our response below for how we address this issue in the manuscript.

In what follows we provide our point-by-point attempt to address the reviewers’ questions.

1) Alternative accounts of these results.The authors consider the possibility that unsigned reward prediction errors experienced during the "relearning" phase prompt revaluation. This is plausible, but the authors don't consider any alternative hypotheses. In the absence of an alternative, it's hard to evaluate whether the evidence they consider in favor of this model is actually decisive – i.e., does their evidence uniquely favor their model, or is it consistent with other models?

As discussed above, we hope that the new manuscript does a better job situating our results within the space of possible accounts. We would first note that the reviewers’ comments in this section appear to center around the interpretation of our secondary (and admittedly more exploratory) result, about which events predict replay. As mentioned above and elaborated below, we now discuss alternative accounts of our results in the fifth paragraph of the Discussion, involving the possibility that a different quantity (e.g. signed rather than unsigned surprise) or surprise at a different time drives replay.

As for our primary result, on the relationship between replay and behavioral change, the only substantive alternative hypothesis we know of here is that behavioral change might also be produced by computations accomplished online. As discussed in the manuscript, these mechanisms are not mutually exclusive, but they do not by themselves explain the results we report.

Moreover, they undercut their own model when they write: "the inclusion of reward noise in the noisy-rewards condition (even in control blocks) allowed us to rule out the possibility that replanning is brought about by 'any' experience of uncertainty, whether or not replanning was optimal". This is a good point and an elegant feature of their design, but they don't seem to address how their model of revaluation assesses the "optimality" of replanning beyond unsigned PEs. In other words, they include a control condition specifically to show that their model is incomplete, and then don't address how it could be made complete.

We have revised the quoted section about “optimality” in the text. We agree with the reviewers that surprise is an imperfect signal as to whether effective replanning is possible: it is, in effect, a necessary but not sufficient condition, since prediction errors may just reflect noise. (This point also comes up later in the review.) As we now say in the paper, to some extent this might be mitigated by, for instance, more carefully attempting to track variance/uncertainty so as to recognize irreducible stochasticity. But the big picture here is that there is no perfect heuristic: it’s not really possible to decide with certainty whether planning will succeed without actually doing it, so we don’t think the theory is “incomplete” in this sense.

Also, we have tried to be clearer about our hypothesis and our interpretation of the noisy rewards condition. As noted above, our hypothesis is that Stage 2 unsigned PEs prompt later *replay* – but this will not necessarily result in effective *replanning* (meaning, changed behavior at Stage 1). Whether or not this replay causes revaluation will depend on whether the world has changed (and hence there is something new to learn from replay). Putting these pieces together, we expected there to be a link between stage 2 unsigned PEs and replanning in the revaluation condition (where the reward changed during relearning) but not in the control condition (even under noise, where the reward structure did not change during relearning). This is what we found, and this pattern of results speaks against alternatives in which, for instance, neural and behavioral signatures of decision variables and replanning covary more generally due to nonspecific changes in, for instance, motivation or attention.

See the Discussion:

“It is important to consider alternative hypotheses and interpretations of our results. […] Altogether, the specificity of the findings is consistent with the hypothesis that learning from simulated experience during replay determines whether revaluation occurs.”

Also see Results:

“This noise manipulation – i.e. the noiseless-rewards versus noisy-rewards conditions – ensured that the difference between revaluation and control condition blocks was not limited to the absence or presence of PEs or surprise in general. […] Accordingly, the inclusion of reward noise in the noisy-rewards condition (even in control blocks) allowed us to rule out the possibility that change in choice preferences is brought about by ‘any’ experience of uncertainty, whether or not experience actually supported replanning.”

Just to sharpen this point, here is one alternative account of what prompts revaluation. Possibly, during the rest period, people start engaging in value iteration from random points in their task model, which includes Stage 1. They then continue the process of value iteration at all points where value iteration results in high unsigned PEs during update. In other words, when performing value iteration on Stage 1 actions, the mismatch between the existing Stage 1 value estimates and the newly updated Stage 2 value estimates generates high unsigned PEs. These remain high until sufficient value iteration occurs that Stage 1 actions are appropriately revalued.In summary, perhaps what drives the total amount of revaluation is not the magnitude of Stage 2 PEs during online relearning, but rather the magnitude (or persistence) of Stage 1 PEs during offline revaluation. (It should be clear how this stands in contrast to the "tagging" idea presented in the general Discussion. Incidentally, there is no reason that both things couldn't happen).[…] The purpose of detailing this alternative is not to argue that this model is right or the only alternative, but to illustrate a much broader point. By sketching only one model of what prompts revaluation and failing to explicitly consider alternatives, the authors leave the reader in a poor position to evaluate whether the analyses that are consistent with their model are in fact good evidence for it – we don't know whether those analyses are also consistent with all plausible alternatives. Could the authors show us what models of revaluation-prompting are ruled out by their analyses? This would be a big help, whether or not the authors decide to discuss the specific alternative mentioned here.

We thank the reviewers for proposing this alternative hypothesis. In general, we view this as a friendly amendment, in that it actually shares the core computational claim of our favored (“tagging”) hypothesis: the idea that surprise triggers replay. The only difference is whether it is surprise during *relearning* or surprise during *replay* that boosts (subsequent) replay. We agree that the data do not discriminate between these two (highly related) accounts and we have updated the manuscript to reflect this. We also note in the paper that there are a priori computational reasons to favor the tagging hypothesis over the random-replay hypothesis: for tasks with larger trees and longer trajectories the random strategy is not tenable. Prioritized replay might seem equally effective in small state spaces, like the one in the present paradigm, but it outperforms the random replay strategy for large state spaces, and especially realistic state spaces with high dimensionality. We have now edited the Discussion to reflect these thoughts:

“Our second key result concerns a further pair of associations, between earlier signatures of PE and subsequent replay and behavioral planning. […] Thus, while acknowledging other possibilities, we suggest that some form of prioritized replay is more likely.”

2) Alternatives raised by the design and logicAkin to the above point, there were several points regarding the design that raise other alternatives, including basic issues of engagement. Specifically:- What incentive did people have to go to the higher-valued states in the learning phase? It seems this part of the task was not incentivized, so people had no particular reason to select higher numbers over lower numbers (except for learning for later). The replanning magnitude metric then becomes potentially problematic in that it subtracts away performance during the learning phase, which was not incentivized.

Thank you for this question. We have added the following to the revised manuscript:

“Participants were incentivized to search for and choose higher-value states. […] And the more money you steal in the game, the more REAL BONUS MONEY you will be paid on top of the normal experiment payment. You could earn an extra $2-3!”

This issue becomes especially problematic when trying to assess the impact of the brain replay (e.g., Figure 2). Here replay correlates with replanning, but which aspect of the replanning is not clear. Perhaps this correlation is due to behaviour in the learning phases-i.e., those people who paid more attention and learned more initially show more replay later on. That's not so surprising or interesting. The interesting thing is to predict later behaviour, but subtracting away a baseline that precedes the replay means that the replay can just be correlating with something that occurred earlier (learning/attention/task engagement, not replanning). It would be more convincing to show that the replay correlates solely with the test behaviour not with this replanning metric.

To be clear, our hypotheses do *not* concern raw test phase performance; the prediction of the theory (as expressed in the DYNA learning equations) is that *change* in the behavioral policy (not the absolute endpoint policy) is driven by replay. Relatedly, and in response to the reviewer’s particular concern, we think it is important to test a replanning score rather than raw test phase performance precisely to control for initial performance during the learning phase. Otherwise, a positive correlation between raw test phase performance and neural activity could in principle be driven by a relationship between the policy learned initially (carried over unchanged into the test phase) and the neural measure. Previous behavioral work based on similar logic (Gershman et al., 2014) used a relative revaluation score for the same reason. We add the following to our explanation of the replanning score:

“We use a score that indexes *change* in preferences here (Gershman et al., 2014), rather than raw Test phase choice behavior for two reasons. First, this is what our hypotheses concern: that replay updates pre-existing choice policies, driving a change. Second, this score controls for any confounding effects of initial learning performance on Test phase behavior. (For instance, subjects paying different amounts of attention might learn to varying degrees in the initial phases, leading to differences in both replay and Test phase behavior, even if replay itself doesn't affect later preferences.)”

Secondly, we would repeat the point made earlier, that our use of the control condition helps to rule out concerns of this sort, since incidental relationships between replay and behavior should be the same in this condition if they are actually driven by attention, motivation, forgetting or other nonspecific factors. Instead we report that the relationship is significantly attenuated in this case, speaking to the specificity of the relationship to the hypothesized condition.

Nevertheless, we examined learning phase behavior separately to verify that participants’ learning-phase behavior was not by itself driving the correlation, we computed a correlation between the participant’s performance on the 10 last trials of the learning phase (i.e. the prescore) and replay and revaluation scores later on. Recall that the key result was a brain-behavior correlation specific to the revaluation condition. But in the revaluation condition (red plots in Author response image 1) there were no positive correlations between replay and learning phase behavior (Revaluation, no Noise: *rho=.19, p=.38*; Revaluation, Noisy: *rho=.34, p=.1*):

We also did not find a significant correlation in the control condition with no noise (*rho = -0.16, p = 0.47*). The only just significant correlation we saw was in the noisy control condition (grey plots in Author response image 2), where we observed a marginally significant negative correlation between prescores (learning phase behavior) and replay (*Spearman’s Rho = -0.41, p = 0.045*). This indicates that lower performance was correlated with more Stage 1 replay during subsequent rest, but this could be also due to uncertainty and does not confound our main results.

**Author response image 2. respfig2:** 

- The above point is compounded somewhat with the choice of the control condition here. Unlike the revaluation condition, there is no learning necessary at all in the second phase. There is also no incentive to pay attention or be engaged at all during this second phase. As a result, the differences in brain activity may be due to differential engagement, rather than specific to the replay-replanning connection as claimed here.

In the revision, we have clarified that attention was required during the relearning phase, for in both the revaluation condition and the control condition. To address this, we have added a paragraph to the main text:

“Attention engagement during the relearning (control trials)Participants were instructed at the beginning of each run that the rewarding choices may change and they needed to pay attention to changes. […] They did not know they were in the control condition and were incentivized to pay attention.”

- The test phase gave no feedback or guidance. How were participants supposed to know which experience to draw upon when making those decisions? The assumption here is that the optimal thing here is to integrate the information in the two phases by adjusting the values of the terminal states in phases 2 and keeping the same phase 1 knowledge of task structure. A smart participant, however, might infer that the recurrence of the Stage 1 stimulus (which never appeared in the relearning phase) meant that the Learning phase reward-contingency rules were back in effect. It's not obvious what is optimal here. Maybe they have learned that the Stage 2 stimuli give different values when they are (or are not) preceded by the Stage 1 stimuli (i.e., a type of occasion setting).

We agree that it is theoretically possible that participants would learn two separate sets of reward-contingency rules in the revaluation condition (one for learning, one for relearning) and then access the initial set of rules during the final test. However, empirically speaking, this account is not correct – it predicts that participants would not show a behavioral replanning effect in the revaluation condition, but they did.

- What were the means/variances of the experienced distributions? Were the actual means of the experienced distributions controlled or only the generative process?

In order to address the reviewer’s first question, we have generated subject-level scatter plots of mean generative reward (x-axis) vs. the rewards the participant observed (y-axis). In response to the second question: we did not fix the realized distribution of rewards but instead generated outcomes pseudorandomly according to the generative distributions. That said, we ensured that the distributions were adequately sampled during the experiment by including forced choice trials. Each participant’s choices were tracked, and actions that were less explored were presented in forced-choice trials in which the participant had to take those actions. We included on average, 3 “forced choice” catch trials in phase 1 of each run. We included at least 1 of each less explored option (out of 4 total options to explore). As a result of this forced-choice manipulation, they sampled all options. In the no-noise conditions this of course ensured that they had repeated experience with the exact means. In the noisy conditions the precise means could slightly vary across participants because the number of visits to each option and their random realizations were not matched across all participants. That said, the rewards were distributed such that this potential variation in the experienced means of the noisy conditions did not change the globally highest mean reward. Therefore, the optimal policy in the global structure remained the same.

**Author response image 3. respfig3:** 

Were the experienced distributions close to the planned ones? How was it ensured that all conditions were experienced? Could people have avoided one of the outcomes e.g., due to getting initially unlucky (hot-stove effect)?

That is an important point that we have now added to the manuscript (subsection “Forced choice catch stimuli”).

In short, we took the possibility of avoiding outcomes into consideration when designing the experiment. To address this possibility, we included controlled *forced choice* trials in which one of the choices on the screen was highlighted and participants had been instructed to choose the highlighted option each time they encountered one. These control forced choices ensured that they explored every branch, and served a number of purposes. (a) They matched the encounters of the different options to ensure participants had exposure to the different conditions. This ensured that the experienced distributions were close to the planned ones, although there were fewer negative PEs, as participants are less likely to return to a choice with negative PE. (b) As mentioned in our response above, these forced choice trials served to keep participants’ attention engaged, as they were told that it is important to choose the highlighted action when an option is highlighted.

If the reviewers see fit, we would be happy to include this in the main text.

- It was not clear is why non-specific replaying the Stage 1 stimuli during the replanning task would help. In some ways, it says that people who are more engaged with the task do better on the task. That's interesting, but not groundbreaking. The data here do not seem to show that category-specific replay predicts later performance. Does replay of other categories (not the Stage 1 stimulus) not correlate with replanning behaviour? That would be a good additional piece of evidence beyond simple task engagement.

We are grateful for the comment. To be clear, we do think that the result concerning Stage 1 stimuli is rather groundbreaking. Importantly, unlike other related work like that of Shohamy and Wimmer, 2012, and Doll et al., 2015, this is activity during a *rest period*: it is thus a bit of a stretch to describe it as reflecting “task engagement.” Perhaps one might still find it unsurprising that thinking about the task even when resting promotes better on-task performance, but this has not to our knowledge been shown previously, and indeed as we discuss in the paper the predominant hypothesis going back to Tolman has been that this sort of task is solved *not* by this sort of off-line contemplation but by on-line planning at choice time, corresponding to our test phase.

Second, our general perspective is that the Stage 1 stimulus (rather than the Stage 2 stimulus) is the most diagnostic signal of the sort of offline replay with hypothesize. The Stage 2 stimulus may also be involved in replay for planning. However, since the Stage 2 stimuli were also visible onscreen during the relearning phase immediately preceding the rest periods where we measure replay, there is likely to be Stage-2 related activity for all sorts of reasons not specific to planning, like simple working memory for recent event. Instead, retrieval of memory for the Stage-1 category (which was not presented in the relearning phase) is the crucial step hypothesized for binding the new information with the Stage 1 actions, i.e. for planning. (We have used similar logic in other studies from our lab, e.g., Gershman et al., 2013, Journal of Neuroscience and Manning et al., 2016, Psychonomic Bulletin and Review.) Overall, we expect the relationship between replay and revaluation to be most clearly expressed for the Stage 1 stimulus.

In light of the reviewer’s comment, we examined the category specificity of the relationship between rest-period category activity and replanning behavior. As expected, the relationship for the Stage-2 stimulus was weaker and not significant. Specifically, we tested whether Stage 2 replay was also correlated with subsequent planning (revaluation) behavior, and we found that the correlation was non-significant for both conditions (revaluation: *rho=.2, p=.17*; control: *rho=.25, p=.22).* Multiple regression did not reveal any significant effect of Stage 2 replay (ß=.84, p=.44) but only a significant effect of the Stage 1 replay on the revaluation scores (ß=1.89, p=.04).

**Author response image 4. respfig4:** 

The relatively stronger effect for Stage-1 rather than Stage-2 stimuli also again suggests that our neural effects are driven by the specific planning-related computations we hypothesize, rather than more generic attention-related confounds.

To address these points in the paper, we have made the following changes.

We added the following to the revised manuscript:

“Note that the crucial step for revaluation is retrieval of memory for the Stage-1 state, which was not presented in the Relearning phase. […] As expected, we observed no significant correlation between Stage 2 replay and revaluation behavior (Figure 2—figure supplement 2) in the revaluation condition (*rho=.2, p=.17*) or the control condition (*rho=.25, p=.22).*”

And included figure 2—figure supplement 2 with the following caption:

“Figure 2—figure supplement 2. We tested whether Stage 2 replay was also correlated with subsequent planning (revaluation) behavior. We found no significant correlation. […] Multiple regression did not reveal any significant effect of Stage 2 replay (ß=.84, p=.44) but only a significant effect of the Stage 1 replay on the revaluation scores (ß=1.89, p=.04).”

3) Concerns over the robustness of the analysis and approach to the computational modeling. It should be noted in light of the first comment below that eLife does not require that all hypothesis, predictions, or analyses be preregistered. However, the reviewers agreed that there should be explicit discussion of investigator degrees of freedom and what safeguards against flexibility in the analysis were put in place. Further, any checks that can be provided for robustness would be helpful. Here are the specific comments raised regarding the analysis and modeling.- The article is pitched as confirmatory in the sense that it makes predictions and then confirms those predictions with evidence. Were any of these predictions pre-registered anywhere? Were there any competing hypotheses? What sort of data might have gone against these predictions? The evidence in the paper could use to be bolstered to establish that it was truly a confirmatory study. Similarly, were any of the metrics or analyses preregistered? If they were indeed hypothesis-driven, that should have been possible. It would seem that there are many, many ways that these data could have been analyzed (both behavioural and neural), and some of the metrics seem very arbitrary (e.g., number of trials in learning phase). Were other analysis pathways attempted? Evaluating the import of the inferential statistics requires more knowledge of the researcher degrees of freedom.

While we did not preregister any of the predictions or analyses/metric, our main predictions were derived from well-established computational models that preceded our experiment by decades, and our analyses were indeed envisioned ahead of time based on other studies that we have conducted in our lab and additional studies from the literature. As discussed in the paper, our prediction linking replay to replanning comes from Sutton’s Dyna architecture (Sutton, 1991), in which action values are updated via offline replay. Our prediction linking surprise during learning to replay comes from prioritized sweeping algorithms (Moore and Atkeson, 1993; Peng and Williams, 1993). The combination of the two was again proposed in Sutton et al., 2012 where the authors combined Dyna-style planning with prioritized sweeping. With respect to the experimental and analytic approaches, Gershman et al., 2014, had tested a similarly structured behavioral task (and analyzed it according to a comparative revaluation index) and Momennejad et al., 2017, use similar reward revaluation paradigms and show that human behavior is consistent with the behavior of an algorithm that employs offline replay. The logic of testing across-subject variation in revaluation success against neural reinstatement measures was derived from Wimmer and Shohamy, 2012, and, in our group, Doll et al., 2015. Backed by these theoretical proposals in reinforcement learning and previous findings in humans, we hypothesized that the brain employs algorithms that use offline replay to update planning policies offline, and that trajectories related to states with local PE would have prioritized replay. Our fMRI replay analyses were based closely on other studies from our lab that used scene/face MVPA to track thoughts related to stimuli not presently onscreen (Gershman et al., 2013; Manning et al., 2016). The method for testing the hypothesis that |PE| drives offline replay and replanning was tested using the common approach in the RL-fMRI literature, which is to estimate the prediction errors using a learning rate from the model, and then use these trial by trial prediction error estimates in a parametric modulation analysis.

Our main predictions could easily have been falsified if our MVPA metric of Stage 1 replay had failed to show a correlation with behavior, or if we had failed to observe a relationship between neural sensitivity to prediction errors and replay. These primary analyses happened to work “out of the box” with minimal adjustment of parameters on our part. Also, as described throughout this response, we have run several control analyses that each could have cast doubt on our interpretation if they turned out a particular way (but they did not). We would be happy to share the data and analyses after publication.

Finally we note that we ran this study in summer 2016 and our lab did not “flip the switch” on preregistering our fMRI studies until 2017. A lot has changed with regard to the prevalence of preregistration in the last 2 years!

- Two participants were excluded post-hoc due to "not having sufficient number of runs". Were there any pre-defined criteria on this front? Or was this exclusion made after the data were examined? Additionally, the sample size seems small (though not inordinately so for an fMRI study), leaving many of the statistical tests underpowered (and indeed many of the brain-behaviour correlations fell in the marginal zone, making interpretation difficult).

We could only include participants who successfully completed trials in all four experimental conditions, and there was 1 run per condition. Thus, participants were excluded on a priori grounds if they failed to learn the initial MDP in any of the four runs. We have now included the following lines to clarify this point in the manuscript:

“Please note that, logically, the analysis required each participants to learn the local dependencies of the task in the learning phase (indexed by their choice behavior) and there was 1 run per condition. […] Therefore, based on this *a priori* requirement, any participant who did not learn the task structure during the learning phase (for any of the four runs) was excluded."

- The idea that high prediction error should be preferentially replayed seems like a rule that would only work in very narrow (static) environments. If there is any variance in the rewards, then this rule could lead to perverse replay of unreducible uncertainty.

That is a great comment and we appreciate that the reviewer brought this up. As also discussed earlier, we agree that in a more sophisticated model tracking variances of various sorts, priority could be scaled to down-prioritize replaying irreducibly uncertain values. (Though we also note that given that part of the goal of replay is to average out stochasticity in computing net action values we don’t actually think it’s entirely pointless to replay those either.) We have now included the following lines to the manuscript to address this point.

“The effect of volatility on prediction errorsHere we have estimated prediction errors regardless of the noise conditions. […] Such a study would benefit from a design where prediction errors are parametrically changed across conditions or groups.”

- The selection of learning rate requires further justification. How was the.7 estimated? What sort of task? If the task were as simple as the current task, with such a wide gap in rewards, I wonder how reliable an estimate this number represents. Moreover,.7 seems extremely high for a task with variable rewards. That would make people extremely influenced by the most recent outcome, much more so than a typical task with variable or probabilistic rewards. Is it reasonable to use the learning rate estimated in a different sample for analyzing the new sample? How robust are the prediction error analysis to this selection of learning rate?

We would like to first stress that while the learning rate sounds high, it is actually pretty typical in our experience for human performance in tasks with stochastic reward. For instance, the learning rates we have estimated in two-step MDPs across many samples exceed 0.5. Here, we estimated this learning rate based on a separate sample of subjects who performed the same task in a behavioral pilot study. The structure (and rewards) used in this pilot study were virtually identical to the fMRI task. As model-based fMRI analyses are typically performed using group-level (rather than individual-level) parameter estimates, we estimated this learning rate from the pooled subject data using Maximum Likelihood Estimation, resulting in one best-fitting learning rate for the entire population. Though this learning rate was estimated from a different sample (a behavioral pilot study), we confirmed, based on this comment, a similar best-fitting learning rate in the fMRI sample (0.78). In this regard, we can confidently say that this learning rate generalizes well across samples. With respect to the generalizability of the neural results to other values of the learning rate, it is worth pointing out that a previous systematic evaluation of this issue found that substantive differences in learning rates in the model-based analysis of similar tasks lead to only minute changes in the neural results (Wilson and Niv, 2015). We have now included the following text in the manuscript to address this concern:

“Estimating learning ratesWe estimated learning rates from a separate sample that performed the same task in a pilot study. […] With regard to the generalizability of the neural results to different values of the learning rate, it is worth noting that a previous systematic evaluation of this issue (Wilson and Niv 2015) found that substantive differences in learning rates in the model-based analysis of similar tasks lead to only minute changes in neural results.”